# Cross-stress gene expression atlas of *Marchantia polymorpha* reveals the hierarchy and regulatory principles of abiotic stress responses

Qiao Wen Tan [1], Peng Ken Lim [1], Zhong Chen[2], Asher Pasha [3], Nicholas Provart [3], Marius Arend [4,5], Zoran Nikoloski [4,5] & Marek Mutwil [1] ✉

Abiotic stresses negatively impact ecosystems and the yield of crops, and climate change will increase their frequency and intensity. Despite progress in understanding how plants respond to individual stresses, our knowledge of plant acclimatization to combined stresses typically occurring in nature is still lacking. Here, we used a plant with minimal regulatory network redundancy, *Marchantia polymorpha*, to study how seven abiotic stresses, alone and in 19 pairwise combinations, affect the phenotype, gene expression, and activity of cellular pathways. While the transcriptomic responses show a conserved differential gene expression between *Arabidopsis* and *Marchantia*, we also observe a strong functional and transcriptional divergence between the two species. The reconstructed high-confidence gene regulatory network demonstrates that the response to specific stresses dominates those of others by relying on a large ensemble of transcription factors. We also show that a regression model could accurately predict the gene expression under combined stresses, indicating that *Marchantia* performs arithmetic multiplication to respond to multiple stresses. Lastly, two online resources (https://conekt. plant.tools and http://bar.utoronto.ca/efp_marchantia/cgi-bin/efpWeb.cgi) are provided to facilitate the study of gene expression in *Marchantia* exposed to abiotic stresses.

The colonization of land by plants, which occurred around 470 Ma, was essential to establish habitable environments on land for all kingdoms of life[1]. Bryophytes, which include mosses, liverworts, and hornworts, represent the earliest diverging group of non-vascular land plants[2–4]. Morphology of the earliest land plant fossils, consisting primarily of tissue fragments and spores from the Middle Ordovician around 470 Ma, showed that early land plants were liverwort-like[5,6]. As a liverwort, *Marchantia polymorpha* is a valuable model to study the emergence and evolution of land plants, as it allows us to compare the biology of aquatic algae and non-vascular plants to vascular, seed, and flowering plants. Studying *Marchantia* can help us better understand the successful terrestrialization event, as *Marchantia* contains traits

[1]School of Biological Sciences, Nanyang Technological University, 60 Nanyang Drive, Singapore 637551, Singapore. [2]Amoeba Education Hub, 1 West Coast Road, 128020 Singapore, Singapore. [3]Department of Cell and Systems Biology/Centre for the Analysis of Genome Evolution and Function, University of Toronto, Toronto, ON M5S 3B2, Canada. [4]Bioinformatics, Institute of Biochemistry and Biology, University of Potsdam, 14476 Potsdam, Germany. [5]Systems Biology and Mathematical Modeling, Max Planck Institute of Molecular Plant Physiology, 14476 Potsdam, Germany. ✉e-mail: mutwil@ntu.edu.sg

essential for this task along with increased complexity (e.g., hormones auxin, jasmonate, salicylic acid, protection mechanisms against desiccation, photooxidative damage)[7] which is still considerably lower than that of vascular plants[8].

Besides its interesting evolutionary position among land plants, *Marchantia* is a valuable model for studying basic plant biology. Mainly due to the lack of whole-genome duplications in the liverwort lineage, *Marchantia* shows a simpler, low-redundancy regulatory genome[8], which together with the ease of growth and genetic manipulation[9], makes *Marchantia* an excellent model to study general plant biology. The *Marchantia* genome contains necessary components for most land-plant signaling pathways with low redundancy, making it easier to dissect the pathways[8]. For example, the auxin signaling network in *Marchantia* is simple yet functional, with all relevant genes existing as single orthologs[10]. Similarly, cellulose biosynthesis in *Marchantia* uses the same but simplified machinery; while *Arabidopsis thaliana* contains 10 cellulose synthases in multimeric complexes, *Marchantia* has only two[11]. Furthermore, since the dominant generation of *Marchantia* is the haploid gametophyte, heterozygosity can be circumvented to directly study mutant and transgenic phenotypes. These, and other features, make *Marchantia* an attractive model for dissecting the function of genes and biological pathways.

Extreme environmental conditions can cause a multitude of biotic and abiotic stresses that can devastate crops and induce the collapse of entire ecosystems[12,13]. Plants have evolved sophisticated mechanisms to perceive and respond to the different stresses, which induces an acclimation process that allows the plant to survive the stress[14], but often at the cost of reduced growth[12,13]. Many studies have analyzed the effect of stress on plant growth by identifying differentially expressed genes (DEGs) between stress-treated and untreated plants (e.g. refs. [15–17]), or by identifying single nucleotide polymorphisms associated with stress resistance in *Arabidopsis* and maize[18–20]. The studies performed on model plants such as *Arabidopsis* can shed light on fundamental processes of stress acclimation, but it is currently unclear whether the acclimation mechanisms are conserved and transferable to crops.

In addition, plants are often exposed to a combination of stresses in the natural environment which may require opposing strategies to mitigate adverse effects. For example, drought causes plants to close their stomata to minimize water loss[21,22], while heat causes the stomata to open to cool down their leaves via transpiration[17,23]. Stress signaling is mediated by a diverse ensemble of stress-specific sensors/receptors, networks of protein kinases/phosphatases, calcium channels/pumps, and transcription factors (TFs) that can be localized to different organelles[24,25]. The propagation of signals is modulated by hormones, signaling molecules (e.g., reactive oxygen species), and other protein modifications (S-nitrosylation, ubiquitination, myristoylation)[26,27]. While this complexity renders it challenging to elucidate the molecular basis of stress responses to single or combined stresses, the key features of the model *Marchantia* provide the means to deepen our understanding of stress acclimation.

In this study, we make use of *Marchantia*'s less complex regulatory architecture to dissect how plants respond to environmental cues, such as stresses, by modulating the expression of genes and biological pathways. To this end, we constructed an abiotic stress gene expression atlas of *Marchantia* comprising seven abiotic stresses, i.e. darkness, high light, cold, heat, nitrogen deficiency, salt, and mannitol, and their 18 pairwise combinations. For each stress, we identified robustly-responding TFs that are likely important for *Marchantia*'s survival to the stress. Comparing these TFs to gene expression responses and biological function of *Arabidopsis thaliana* orthologs revealed poor agreement between the two plants. However, when we looked at the profile of differentially expressed genes in *Marchantia* and *Arabidopsis* homologous gene families, there were significant similarities in cold, heat, and salt stress based on the Jaccard Index (JI), suggesting

conservation of TF responses in *Marchantia* and *Arabidopsis* at the gene family level. Interestingly, the analysis of DEGs and biological pathways in the combined stresses revealed that certain stresses (e.g., darkness and heat) induce large transcriptomic responses that dominate other stresses (e.g., salt and mannitol). The dominant stresses express a larger ensemble of TFs that change the expression of more genes and pathways than the non-dominant stresses. Importantly, we construct a linear regression model that can explain the gene expression changes of combined stresses when employing $\log_2$-fold change ($\log_2$fc) values, showing that *Marchantia* performs an arithmetic multiplication to integrate environmental cues. Finally, to provide bioinformatical resources, we created i) an eFP browser for *Marchantia* (http://bar.utoronto.ca/efp_marchantia/cgi-bin/efpWeb.cgi) that allows the visualization of gene expression in organs and stress conditions and ii) an updated CoNekT platform (https://conekt.plant.tools)[28], allowing sophisticated comparative gene expression and co-expression analyses.

## Results

### Response of *Marchantia* to combined abiotic stresses

To capture gene expression changes caused by a single or combination of stresses, we first established the type and magnitude of stresses to use. We defined two types of stresses: (i) the environmental stresses comprised heat, cold, excess light, and prolonged darkness, while (ii) media stresses comprised nitrogen deficiency, excess salt (representing ionic and osmotic stress), and excess mannitol (representing osmotic/drought stress). Next, the magnitude of the stresses was modulated to identify near-lethal stress conditions, growth decrease by ~50% (inferred from the approximate thallus area), or stresses displaying signs of necrosis. To this end, gemmae grown on sealed, sterile agar plates under constant light were subjected to varying degrees of stress, and their phenotypes were observed on days 15 and 21. For media stresses, the gemmae were subjected to the stress from day 0, while for the environmental stresses, the stress was applied on day 14 for 24 h (Fig. 1a). Prolonged darkness was an exception to this design, as plates were subjected to darkness on days 8, 9, 10, 11, 12, 13, and 14 to expose the plants to 7, 6, 5, 4, 3, 2, and 1 day(s) of darkness at day 15, respectively (Fig. 1a).

Single stresses showed varying degrees of effect on plant growth on day 15 (day of harvest when compared to their respective batch controls, Supplementary Fig. 1 shows growth measurements against respective batch controls, Source Data 1 shows agar plates, Fig. 1d shows growth measurements against controls averaged across all batches) and day 21 (growth measurements for 6 days post-stress for environmental stresses against controls averaged across all batches, Fig. 1e). One day of cold stress did not affect the growth at the temperature range tested (3–12 °C), and we selected 3 °C for further analysis. The heat stress experiment showed that the plants abruptly died when the heat treatment temperature was increased from 33 °C (no phenotype) to 36 °C (death), and we selected 33 °C for further study. For light stress, we selected 435 $\mu$Em$^{-2}$s$^{-1}$ as we observed necrosis at the next higher light intensity (535 $\mu$Em$^{-2}$s$^{-1}$, Source Data 1). For osmotic (100 mM selected) and salt stress (40 mM selected), we observed an expected negative growth gradient when the concentration of the two compounds was increased. For nitrogen deficiency, at 0 mM KNO$_3$, we observed a decrease in growth and an accumulation of a red pigment, which likely represents auronidin, a flavonoid shown to accumulate during phosphate deficiency[29]. Finally, for darkness, we observed that the growth of plants decreased proportionally with the duration of days without light, and we selected plants grown in 3 days of darkness, as they showed a growth decrease of ~50% on day 15 (Supplementary Fig. 1). On day 21 (i.e., 6 days of normal growth condition), all dark-grown plants showed increased size, indicating that the 7 days of darkness is not lethal.

Next, we determined how *Marchantia* responds to a combination of two stresses. To this end, we tested all 19 feasible pairs of stresses

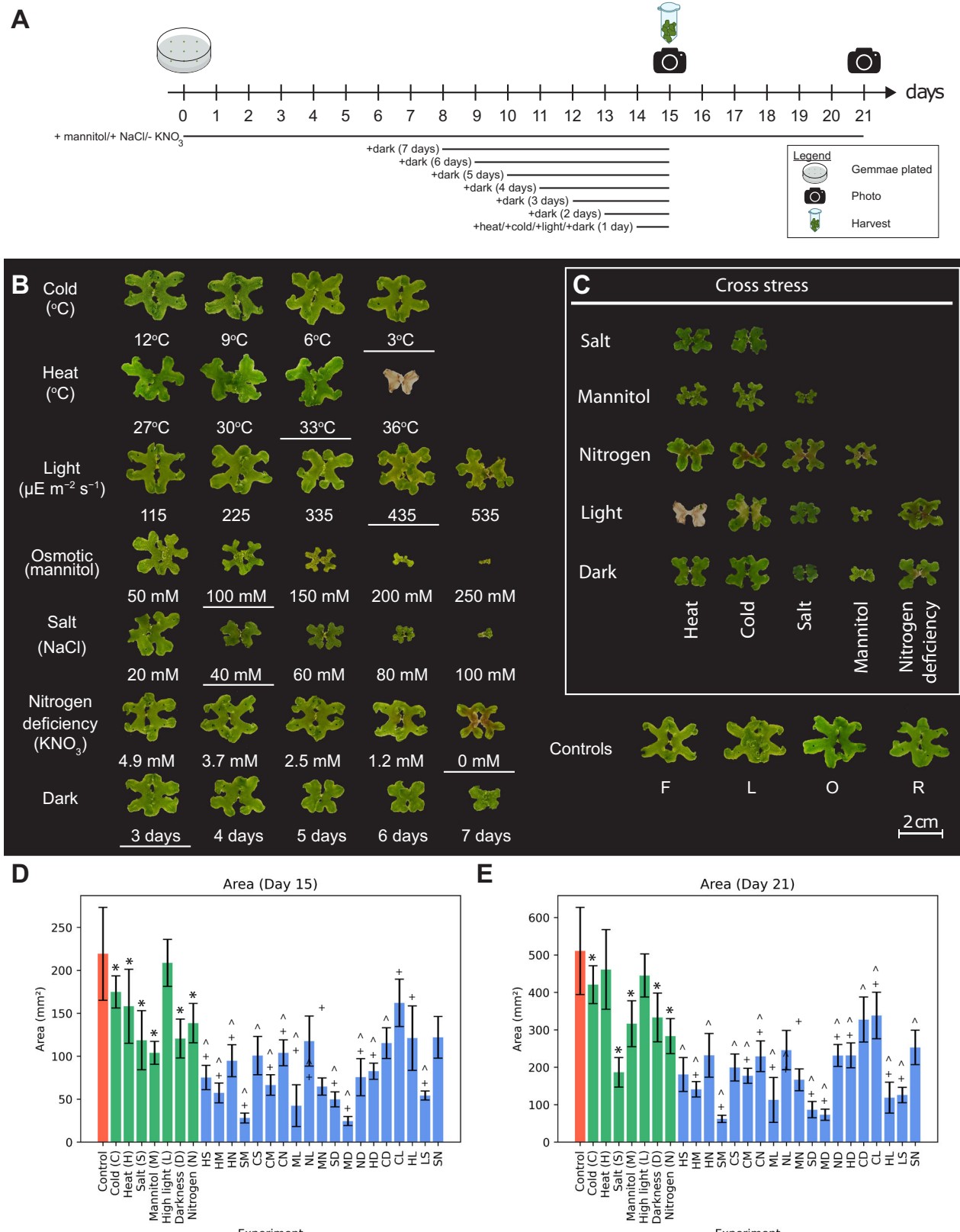

(cold+heat and dark+light combinations are mutually exclusive and excluded) using the same experimental outline as for single stresses (Fig. 1a). We did not observe any unexpected phenotypes when combining the stresses (Fig. 1c), as typically, a combination of stresses resulted in an expected combination of phenotypes (e.g., nitrogen: small, pigmented, mannitol: small, nitrogen + mannitol: even smaller, and pigmented) except for salt-nitrogen (SN), which is significantly larger than its salt counterpart but not different from its nitrogen counterpart (Fig. 1e). While plants subjected to a combination of sub-lethal heat (33 °C) and high-light (435 μEm$^{-2}$ s$^{-1}$) died (Fig. 1c), this was most likely due to a greenhouse effect caused by high irradiation and sealed plates, as the temperature of agar climbed to 38 °C (i.e. lethal

**Fig. 1 | Influence of different abiotic stresses on the growth of *Marchantia*.**
**a** Overview of the experimental setup for stress experiments. Black lines below the timeline indicate the duration where plants were exposed to stress. Plants were sampled on day 15. Cultures for observation were returned to normal growth conditions on day 15, and photographs of the cultures were taken on days 15 and 21. **b** Phenotype of plants on day 21 for heat, cold, osmotic, salt, light, darkness, and nitrogen deficiency stresses. Conditions with underlines represent the stress intensities used for combined stress. Representatives of control from batches F and L for independent stresses performed over 14 batches in panel **b** (Supplementary Fig. 1 and Source Data 1). **c** Phenotypes of plants on day 21 for combined stresses.

Representatives of control from batches O and R for combined stresses performed over 9 batches in panel **c** (Supplementary Fig. 1 and Source Data 1). **d, e** Thallus size of gemmalings on Days 15 and 21. Error bars are represented by standard deviation and Student's two-tailed *t*-test, *p*-value < 0.05. Asterisk (*) represents a significant difference to control, caret (^) represents a significant difference to the first single stress control, and plus sign (+) indicates a significant difference to the second single stress control. The data comprises 131 samples (Control), 8 samples (Cold, Light, Dark), 9 samples (Heat, Salt, Mannitol, HS, HM, HN, SM, ML, NL, MN, SD, MD, ND, HD, CD, CL, HL, LS, SN) and 7 samples (Nitrogen, CS, CM).

temperature, Fig. 1c). Interestingly, we observed yellowing of the thalli for high light and cold, indicating that lowering the temperature makes the plants more sensitive to high light (Fig. 1c).

The resulting panel of 7 single stresses, 18 combinations of two stresses (excluding heat-light that resulted in plant death), and two untreated controls were sent for RNA sequencing (Supplementary Data 1 contains sample labels, and Supplementary Data 2 and Supplementary Data 3 contain Transcript Per Million (TPM) values and raw counts, respectively). Overall, we observed a good agreement between the sample replicates, as samples showed expected clustering (Supplementary Fig. 2), and the correlation between expression profiles of replicates was >0.97 (Supplementary Data 4).

## Combinatorial differential gene expression analysis reveals a hierarchy of stress responses

Plants perceive and respond simultaneously to multiple stresses when growing in nature. To better understand how *Marchantia* responds to stresses, we first identified DEGs in the single stresses and the 18 combinations of two stresses. For more robust inferences, we used two controls (grown on half-strength Gamborg B-5 Basal agar at 24 °C under continuous LED light at 60 µEm$^{-2}$ s$^{-1}$) taken from batches F and L and considered a gene differentially expressed (denoted by DEG) if it showed conserved differential expression with respect to both controls (Supplementary Data 5 and Supplementary Data 6). Overall, we observed a good agreement between the two controls, demonstrated by both the volcano plots (Supplementary Fig. 3) and comparisons of the resulting lists of DEGs (Supplementary Fig. 4A). We compared the number of up- and down-regulated genes with the single stresses and observed that the combination of stresses typically contains a similar or higher number of DEGs when compared to single stresses (Fig. 2a).

Next, to investigate whether a severe growth phenotype results in a larger number of DEGs, we plotted the plant size (*x*-axis) against the number of differentially expressed genes (*y*-axis, Supplementary Fig. 4B). We observed no significant correlation between these two variables for plants on days 15 and 21 (*p*-value > 0.05, Supplementary Fig. 4B). We concluded that there is no correlation between the severity of growth phenotype and the transcriptomic response to stress. For instance, the smallest plants (salt+mannitol, SM) also had the fewest number of differentially expressed genes.

The different single and combined stresses are likely to elicit similar and unique gene expression responses, resulting in the stresses having similar sets of DEGs. We used the UpSet plot[30] to elucidate these similarities, which shows the intersection of multiple sets for upregulated (Supplementary Fig. 5A) and downregulated (Supplementary Fig. 5B) genes. Interestingly, the largest set of up- and down-regulated genes was unique to heat+darkness combined stress (HD), suggesting that HD elicits the most unique and dramatic transcriptional response among the tested stresses. Other unique stress responses comprised upregulated cold+darkness (CD), cold+high light (CL), cold+nitrogen deficiency (CN) and heat+nitrogen deficiency (HN); and down-regulated cold+darkness (CD), cold+nitrogen deficiency (CN), cold (C) and cold+high light (CL). Darkness alone (D) and in combination with

other stresses (e.g., ND, CD, MD, SD, HD) also contained a high number of upregulated (connected dots in columns 4, 5, Supplementary Fig. 5A) and downregulated (columns 2, 4, Supplementary Fig. 5B) genes, suggesting a conserved, core darkness response. Similarly, we observed core responses to heat (e.g., 6th and 10th column, Supplementary Fig. 5A) and cold (11, 12, 19, and 39th column, Supplementary Fig. 5A). We also observed a high number of DEGs across heat and darkness experiments (Supplementary Fig. 5A, B), suggesting that these two stresses elicit a similar response to a degree.

To better understand how *Marchantia* responds to a combination of two stresses, we compared the combined response (Sxy) to the response to individual stresses (Sx and Sy), with three different metrics measuring the shared response, the dominance of stress, and novel responses induced by combined stress. We first produced Venn diagrams for upregulated (Supplementary Fig. 6) and downregulated (Supplementary Fig. 7) gene sets. The first metric measures the similarity between Sx and Sy (Fig. 2b, green area, JI) and ranges from 0 (Sx, Sy do not have any DEGs in common) to 1 (Sx, Sy elicit identical DEGs). The second metric measures whether one stress suppresses the other (Fig. 2b, the difference between red and blue area) and ranges from −1 (Sxy is similar to Sx but not to Sy, i.e., Sx suppresses Sy) to 1 (Sxy is similar to Sy but not Sx, i.e., Sy suppresses Sx). The third metric measures whether a combination of two stresses elicits a unique response when compared to the two individual stresses (Fig. 2b, purple area specific to Sxy) and ranges from 0 (all DEGs are found in Sx and Sy, i.e., no novel response) to 1 (all DEGs in Sxy are unique, i.e., the combination of Sxy elicited a unique response). Using the comparison of upregulated genes in cold+dark as an example (Fig. 2c), the similarity between upregulated genes in cold and dark stresses is low (white field, 0.04). Furthermore, cold+dark combination showed a comparably high proportion of novel interactions (medium blue, 0.37), indicating that the combined cold+dark stress upregulates genes not found in either cold or dark stress. Lastly, the suppression analysis (Fig. 2e) revealed a positive value (light red field, 0.22), indicating that there is a high proportion of genes that are upregulated during cold, but not upregulated in cold +dark. This indicates that darkness suppresses part of the upregulated cold stress response.

Stresses showing the highest similarity in terms of DEG responses comprise salt and mannitol for up- and down-regulated genes (Fig. 2c, d, similarity values for SM is 0.27 and 0.20, respectively), salt and nitrogen deficiency for upregulated genes (Fig. 2c, similarity value 0.21), and heat and darkness for downregulated genes (Fig. 2d, similarity value 0.26). The set analysis for novel interactions revealed that the salt+mannitol combination involved DEGs that were not found in the individual stresses (Fig. 2c, d, novel interaction value 0.48 and 0.56, for up- and downregulated genes), suggesting that the two stresses can activate altogether different responses when combined. Lastly, the suppression analysis showed that darkness is a strong suppressor for many stresses (negative suppression scores ranging from −0.21 to −0.65 when D is Sx, Fig. 2e, f), except HD (−0.09 and −0.06 for up- downregulated genes, Fig. 2e, f), indicating that heat stress and darkness are comparably

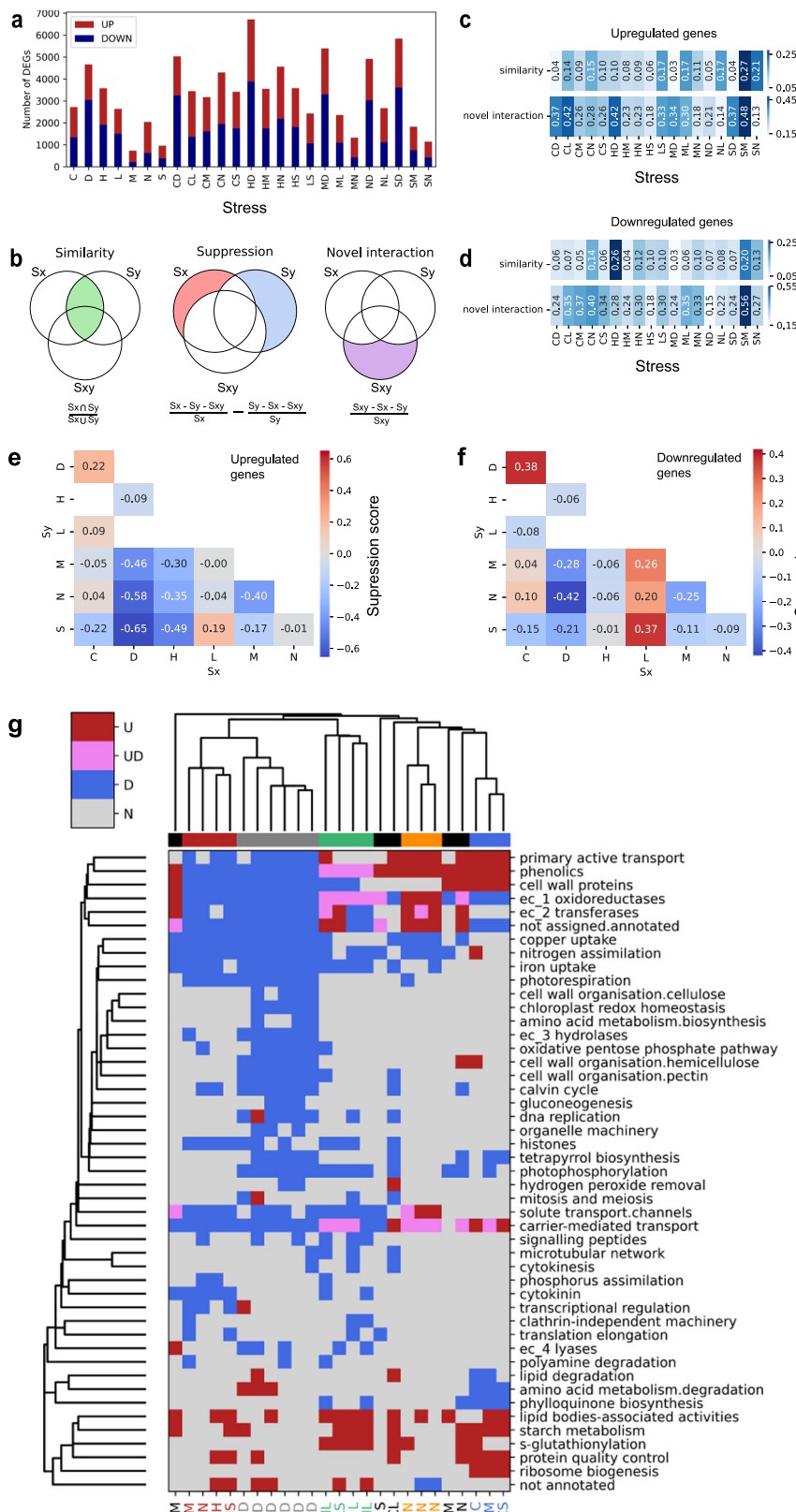

**g**

dominant. To support this observation, we found that heat stress could suppress other stresses (negative suppression scores ranging from −0.01 to −0.49 when H is Sx. Fig. 2e, f).

To better understand the hierarchies of the stresses and how these stresses affect the activity of biological processes, we identified processes that contained significantly more upregulated or

downregulated DEGs than expected by chance for each stress combination (Fig. 2g). The clustering of the rows (biological processes) and columns (stresses) revealed that darkness-containing stresses form a clear group of similar responses (six stress combinations comprising D, HD, CD, MD, SD, and ND), confirming the previous observation of darkness suppressing other stresses. Interestingly, darkness caused a

**Fig. 2 | Analysis of differentially expressed genes and biological pathways.**
**a** The number of significantly (BH-adjusted p-value < 0.05) upregulated (red) and downregulated (blue) differentially expressed genes. The stresses are (C)old, (D) arkness, (H)eat, (L)ight, (M)annitol, (N)itrogen deficiency, and (S)alt. **b** Illustration and equation of metrics used to measure similarity between two stresses (left), suppression of one stress when two stresses are combined (middle), and novel genes that are differentially regulated when stresses are combined (right). Similarity and novel interaction between independent and cross stresses for **c** upregulated genes and **d** downregulated genes. In each stress combination, the first (Sx) and second (Sy) stress corresponds to the first and second letter of the combined (Sxy) stress, respectively. The values were calculated from the equations given in **b**. Suppression analysis for **e** upregulated and **f** downregulated genes. A darker shade of red and blue indicates that more genes from the first (Sx) and second (Sy) stress are not represented in the combined stress (Sxy), respectively. The values were calculated from the equations given in **b**. **g** Biological processes that were significantly differentially expressed (hypergeometric test with 1000 permutations, BH-adjusted p-value < 0.05). For brevity, we only show Mapman bins that were differentially expressed in at least three stress perturbations. The groups of stresses are color-coded. Abbreviations used to describe the categories of regulation are upregulation ('U', red), up and downregulation ('UD', purple), downregulation ('D', blue), and no change ('N', gray).

strong decrease in gene expression of numerous pathways (Fig. 2g, ~58% blue squares). The second largest group contained nearly all heat stress combinations (four stresses: H, HM, HS, HN) and a similar but less dramatic downregulation of transcripts in many biological processes. Interestingly, despite the dramatic downregulation of biological processes in most dark and heat stresses, a subset of stresses (H, HS, HD, MD) was significantly upregulated for uncharacterized genes (bin 'not annotated'), suggesting that these responses employ poorly understood genes. The other groups comprised high-light (four stresses L, NL, LS, ML), nitrogen deficiency (three stresses N, MN, and SN), and cold (three stresses C, CM, and CS). In contrast, salt- and mannitol-containing stresses did not form any groups, suggesting that despite dramatic phenotypic changes, ***these stresses are suppressed by other stresses (Fig. 2g, M and S are not grouped with other stresses). Interestingly, salt+mannitol (SM), cold+high light (CL), and cold+low nitrogen (CN) were also not grouped, indicating that these combinations result in novel transcriptomic responses.

Based on these findings, we can rank the strength of dominance of abiotic stresses starting from darkness (six stress combinations), heat and light (four each), nitrogen deficiency and cold (three each), and finally, salt and mannitol (none).

## Identification of high-confidence transcription factors involved in stress response

Our results indicate that certain stresses (e.g., heat and darkness) result in a large number of DEGs (Fig. 2a). These DEGs are likely a result of the action of a gene regulatory network (GRN) comprising TFs that are themselves differentially expressed.

To infer the stress-responsive GRNs (Supplementary Fig. 8B), we used ElasticNet, to build regression models that can predict gene expression of a gene (response variable), given the expression of transcription factors (predictor variables). We chose ElasticNet, since regularized regressions achieved competitive performance when compared to other GRN inference methods[31], and unlike some of the top performing methods (e.g., Genie3[32]), facilitate easier comparison of inferred interactions across different datasets. We constructed one model for each of the 6257 DEGs (response variable) and all 95 differentially expressed TFs (predictor variables) that were responsive in more than five experiments (Supplementary Fig. 8A). In addition to constructing the GRNs from the whole dataset (i.e., all 81 RNA-seq experiments), we also constructed stress-specific GRNs by using the data from those experiments that included the respective stress (summarized in Supplementary Fig. 8B). For instance, the darkness-specific GRN was inferred from D, DC, DH, DS, DM, and DN expression data. Altogether, we constructed 50,056 ElasticNet models (i.e., 6257 DEGs for eight stress datasets) containing up to 95 differentially expressed TFs as predictors. To study the performance of the models across the different expression datasets, we investigated the number of DEGs (response variables) for which the corresponding models showed coefficient of determination ($R^2$) greater than or equal to a given value. Interestingly, the number of DEGs for which reliable models ($R^2 > 0.8$) could be obtained was larger when using the compilation of experiments that shared a stress, rather than when using all

data (Supplementary Fig. 9A). This suggests that there is significant variability between the datasets that can be used for linear modeling at the stress group level but not when all data sets are jointly examined. As a result, the GRNs based on the individual stress groups showed higher similarity based on the JI to experimentally-derived GRN from *Arabidopsis* than the GRN based on all datasets (Supplementary Fig. 9B). Furthermore, by taking the union of the stress-specific networks, we obtained a GRN with the highest similarity to the *Arabidopsis* GRN (p-value < 0.05, Supplementary Fig. 9B). Thus, we settled on the union of the models from seven stress-specific GRN, with $R^2 > 0.8$ performance. Next, to obtain a high-confidence GRN, we selected the TF with the highest absolute relative coefficient for each gene (Fig. 3a). Based on the value of the selected coefficients, the majority of the inferred relationships reflect TFs as activators (3355 positive coefficients, green edges), followed by repressors (1338 negative coefficients, red edges) and ambiguous (1185 mixture of positive and negative coefficients, gray).

To identify which TFs are robustly responding to a given stress, we visualized the significantly up- and down-regulated TFs across all available combinations of stresses (Fig. 3b). Interestingly, certain TFs show consistent expression patterns across most combinations of a given stress group (e.g., *Mp2g00890.1* is downregulated in 5 out of 6 cold stress combinations, Fig. 3b, bottom row and Supplementary Fig. S10, 26th row from bottom). In total, we identified 75 TFs that showed a consistent, robust response across >70% of combinations within a stress group (termed robustly-responding TFs). The number of robustly-responding TFs in a stress group corresponds to the number of differentially expressed genes. For example, a large number of robustly-responding TFs are found for stresses with a higher number of DEGs (Fig. 3c, darkness, heat), while stresses with few DEGs had fewer robustly-responding TFs (salt, nitrogen deficiency). To study the roles of these 75 robustly-responding TFs, we considered the available literature on NCBI for studies that investigate the molecular function of the TFs. Out of the 75 TFs, 14 were found in the literature and only 3 have been studied for their role in abiotic stress response (Mp*LAXR* (*Mp5g06970.1*, robustly upregulated in high light and mannitol stress) and Mp*ERF15* (*Mp7g09350.1*, robustly downregulated in heat stress): regeneration after wounding, Mp*HSF1* (*Mp4g12230.1*, robustly downregulated in heat stress): heat tolerance) (Supplementary Data 7 and Fig. 3c). The remaining 11 TFs have roles in various developmental processes (e.g., gemma cup formation, Supplementary Data 7), indicating that abiotic stress influences the development of Marchantia.

We expect that the observed downregulation of biological processes in darkness should be caused by upregulation of repressors, downregulation of activators, or both. Interestingly, in darkness, the downregulated TFs comprise mainly of activators (leftmost column, green cells, Fig. 3c). In contrast, the upregulated TFs contained many repressors (red cells), suggesting that the large downregulation of most biological processes is due to the combined action of repressed activators and expressed repressors. Finally, most TFs showed specific expression in at most one stress with few exceptions, such as *Mp8g11560* (robustly downregulated in all stresses) and *Mp4g17430* (upregulated in 5 out of 7 stresses).

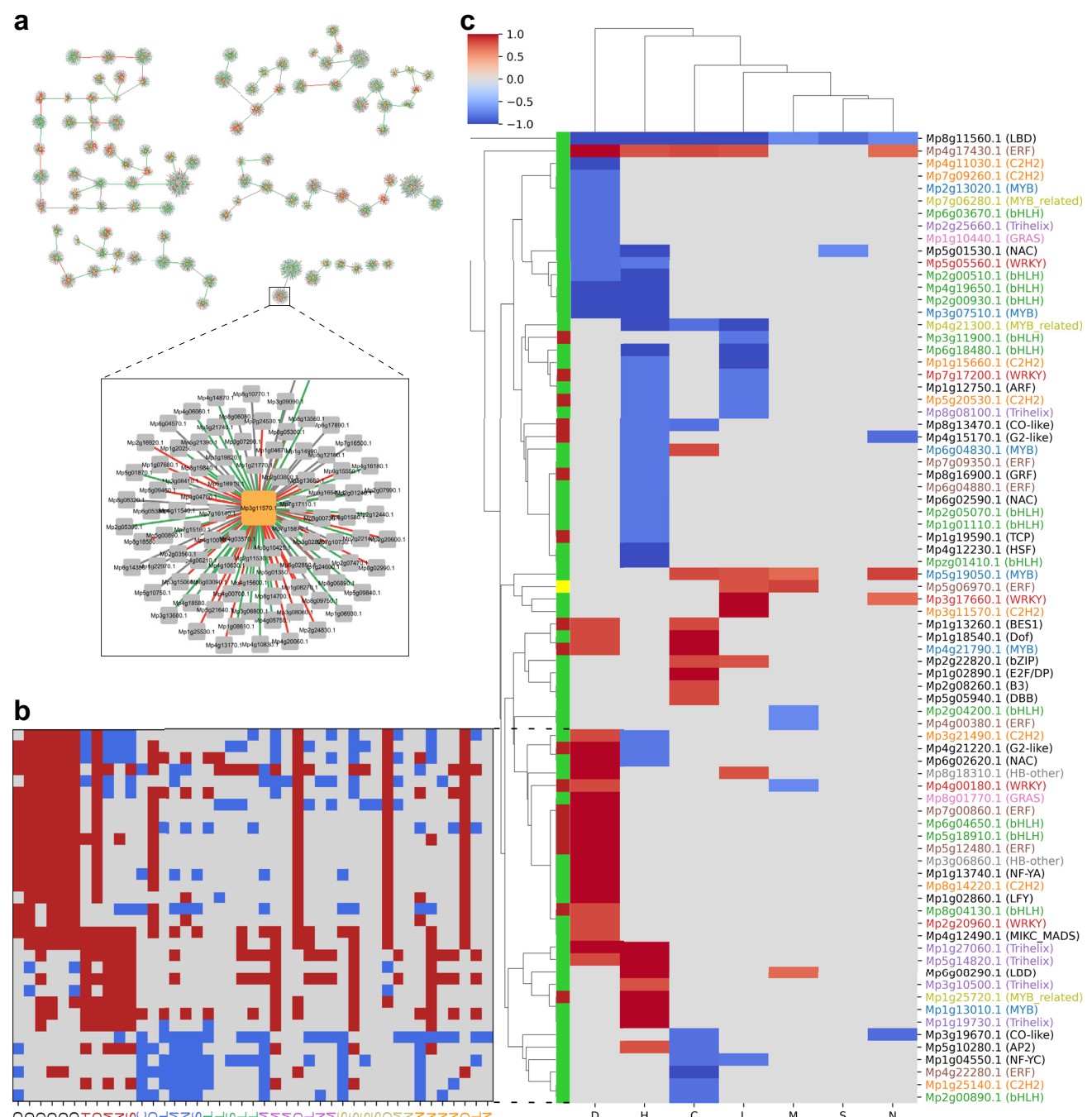

**Fig. 3 | Identification of robustly-responding transcriptional activators and repressors. a** Gene regulatory network constructed from the union of the seven stress-specific networks. For each of the 6257 differentially expressed genes, we kept models with $R^2 > 0.8$ and selected one TF with the highest absolute relative coefficient. Orange and gray nodes represent TFs and genes, respectively, while green and red edges represent positive and negative coefficients, respectively. **b** Differential expression patterns of TFs across stress groups. Red, blue, and gray indicate significantly up-, down-regulated, and unchanged expression, respectively. Stresses are colored according to the stress groups. **c** Identification of 75 robustly-

responding TFs across the stresses. For clarity, cells with specificity scores <0.7 are masked. Red and blue cells indicate the degree of up- and down-regulation, respectively. Green, red, and yellow colors on the leftmost column of the plot represent the most commonly observed relationship between TF and gene, which corresponds to an activator, repressor, and ambiguous (positive and negative coefficients observed for the same TF-gene pair in different stress-specific networks) respectively. TFs are colored according to their TF family for TFs that are represented at least thrice.

## Construction of stress-responsive transcription factor regulatory network

To gain a robust, genome-wide view of the *Marchantia* stress-responsive transcription factor regulatory network (TFRN), we set a global coefficient threshold of the ElasticNet Regression that best explains the observed DEG patterns. To achieve this, we

differentiated 'expected' from 'unexpected' gene regulatory patterns (see "Methods" section). For example, an upregulated transcriptional activator is expected to upregulate its target, and conversely, an upregulated repressor is expected to downregulate its target (Fig. 4a). We then set to identify the coefficient that produced the-highest ratio of expected / total regulatory edges ranging from 0

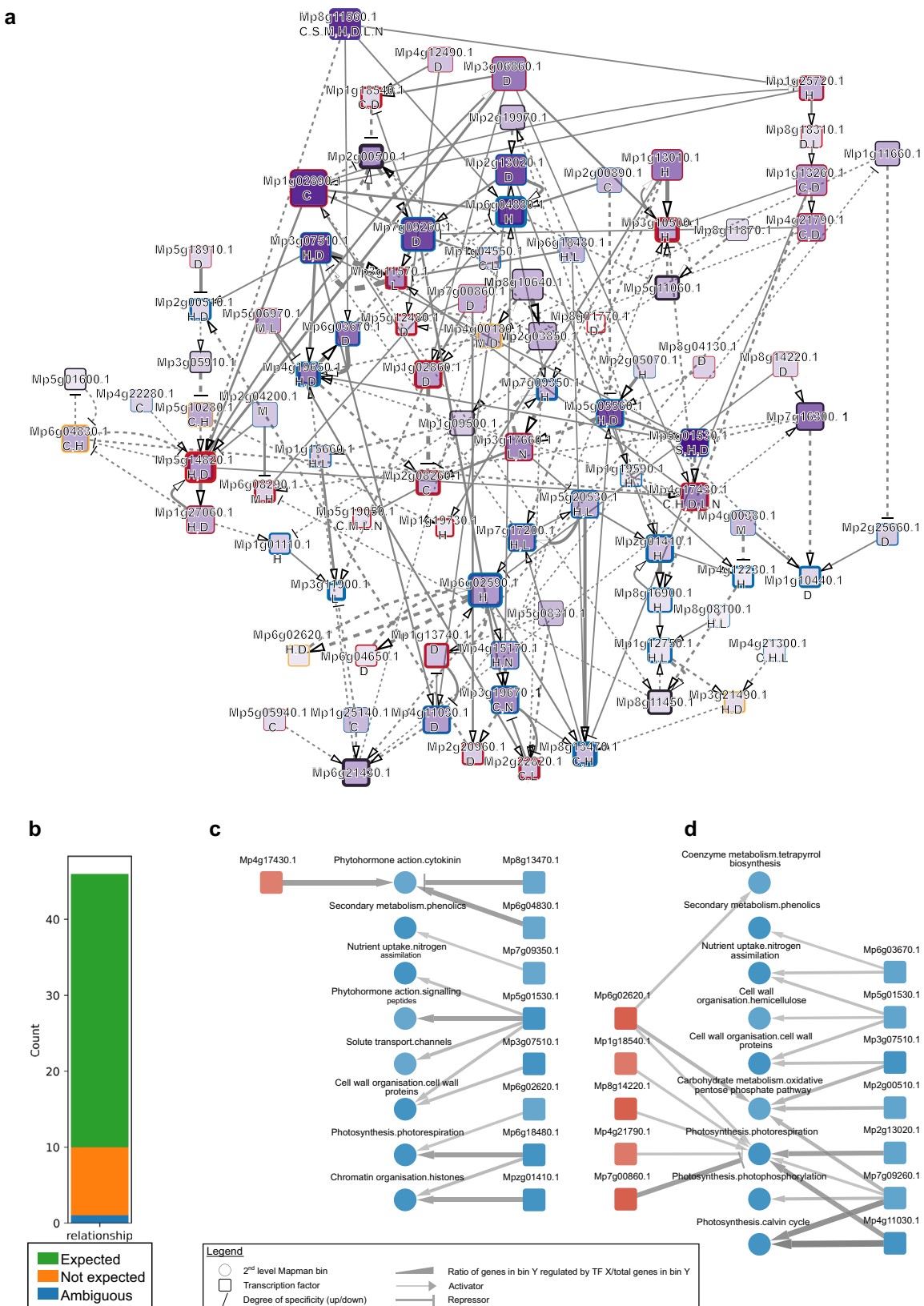

(no expected edges were observed) to 1 (all edges were expected). The analysis revealed that at a coefficient cut-off of 0.22, the highest ratio was achieved (48.4% of edges can be explained, Supplementary Fig. 11A), while at the same time, most (89 out of 95) TFs were still connected to other TFs in the GRN (Supplementary Fig. 11A, red line, Supplementary Fig. 11B).

The resulting TREN revealed intricate regulatory relationships between the 89 TFs. TFs with the highest number of regulatory targets (dark node color) are heat- and dark-related (indicated by H, D in the node name) (Supplementary Fig. 12). At the same time, these TFs also regulate the highest number of other TFs (larger nodes indicate TFs controlling a higher number of other TFs). Interestingly, TFs with the

**Fig. 4 | Analysis of stress-responsive transcription factor regulatory network.** **a** Transcription factor regulatory network comprising of TFs differentially regulated in at least five experiments. Labels below the gene name indicate robustly-responding TFs in (C)old, (D)arkness, (H)eat, (L)ight, (M)annitol, (N)itrogen deficiency, and (S)alt. Pointed and blunt arrows correspond to transcription activators and repressors respectively. Darker node colors indicate a higher number of genes a TF is regulating, while the node sizes indicate the number of TFs the node is regulating. Node border width indicates the number of incoming regulatory signals of the TF. Red, blue, and yellow node border colors indicate TFs that are robustly upregulated, downregulated, or both. Thicker edges indicate higher absolute coefficients, where pointed and blunted arrows represent positive and negative coefficients, respectively. Solid and dashed edges represent expected and unexpected regulations, respectively. **b** The number of expected, unexpected, and ambiguous gene regulatory relationships between TFs and MapMan bins. Only edges between TFs controlling ≥5% of genes in a MapMan bin are used. Identification of TFs that regulate biological processes during **c** heat stress and **d** darkness. Red and blue nodes indicate robustly up- and down-regulated second-level MapMan bins, respectively. Edge thickness represents the percentage of genes controlled by a TF, in a given MapMan bin, with thicker edges indicating a higher percentage. Pointed and blunt arrows indicate that a TF is an activator or repressor of a given process.

highest number of regulatory targets are typically downregulated (dark nodes with blue borders).

## Investigating the regulatory wiring of stress-responsive transcription factors and biological processes

Next, we set out to investigate which biological processes the robustly-responding TFs regulate in the different stresses. We first investigated which biological processes are robustly differentially expressed by finding MapMan bins that show consistent expression pattern changes across the stress combinations (Supplementary Fig. 13). Next, we calculated the percentage of target genes in each MapMan bin that a given TF regulates, based on the above GRN. The number can range from 0 (a TF regulates 0% of genes in a bin) to 1 (a TF regulates 100% of the genes). We set a threshold of 5% target genes in the MapMan bin based on the distribution of the percentages across the network (Supplementary Fig. 14), as at this threshold, the majority of regulatory relationships are expected (e.g., upregulated activator results in an upregulated bin, Fig. 4b), and most TF-MapMan bin edges are removed (Supplementary Fig. 14), resulting in a sparse network. We observed that multiple TFs typically regulate each biological process (Figs. 4c, d and S15). For example, the expression of cell wall proteins is decreased in heat (blue node 'Cell wall organisation.cell wall proteins', Fig. 4c), and this biological process is controlled by two downregulated TFs: *Mp5g01530* and *Mp3g07510* (blue downregulated nodes). At the same time, a TF can regulate multiple biological processes, as exemplified by dark stress-specific downregulated activator *Mp7g09260* downregulating multiple processes related to photosynthesis (Fig. 4d). Thus, the inferred GRN can serve as a resource to dissect how *Marchantia* can cope with abiotic stresses.

## Functional comparison of transcription factor responses in *Marchantia* and *Arabidopsis*

Often, scientists study model organisms with the hopes of understanding the biology of other species. However, it is currently unclear how conserved the GRNs are between species. If *Arabidopsis* and *Marchantia* show a high degree of GRN conservation, we expect the TF orthologs to show conserved expression patterns and be necessary for survival under the same stresses.

To gauge how similar the stress-specific responses are between *Marchantia* and *Arabidopsis*, we identified the *Arabidopsis* orthologs of the 95 stress-responsive *Marchantia* TFs and studied their experimentally-verified biological function and stress-responsive expression patterns. Typically, each *Marchantia* TF has many *Arabidopsis* orthologs due to larger gene families in *Arabidopsis*, and the functions of the *Arabidopsis* orthologs are annotated based on evidence from literature or gene expression from the eFP browser abiotic stress dataset[33] (Supplementary Data 8). To summarize the findings, we grouped the *Marchantia* TFs according to the stress-specific occurrence and counted the number of literature and gene expression evidence in the corresponding *Arabidopsis* orthologs (Fig. 5a and Supplementary Data 9). For most stresses, we did not find visible congruence between the transcriptomic response of *Marchantia* TFs and the biological function and transcriptomic response of the *Arabidopsis* orthologs (Fig. 5a and Supplementary Data 9). For example, the heat-responsive *Marchantia* orthologs in *Arabidopsis* have experimentally-verified biological functions in cold, dark, salt, mannitol/drought, and nitrogen deficiency (Fig. 5a) with the majority of functions not being involved in heat stress (Fig. 5b, <20% *Arabidopsis* orthologs invovled in heat response, first column on the left). We also observed similar patterns for the gene expression responses of *Arabidopsis* orthologs (Fig. 5c), where most observed gene expression responses were not related to heat. To investigate whether the differential gene expression patterns of stress responses are similar between *Marchantia* and *Arabidopsis*, we downloaded the differentially expressed genes for *Arabidopsis* cold, dark, heat, and salt[16]. Next, we calculated the JI similarity of the gene families that are differentially expressed between *Arabidopsis* and *Marchantia* genes. The analysis revealed that for all stresses, with the exception of dark, the number of similar gene expression families that are differentially expressed in the stresses is higher than expected by chance (Fig. 5d, the observed JI is a black dot, significantly higher JI values are indicated by black asterisk). This indicates that despite the seemingly different responses (Fig. 5a–c), the two plants differentially express a similar set of gene families (Fig. 5d).

## Regression-based prediction of stress-responsive gene expression

Our analysis of significantly differentially regulated MapMan bins across experiments revealed that specific stresses (e.g., darkness, heat) can dominate other stresses (e.g., salt, mannitol, Fig. 2g). This indicates that when two stresses ($S_x$ and $S_y$) are present, the combined stress $S_{xy}$ may resemble one of the stresses more than the other. However, the rules governing how gene expression values change in combined stress when two genes are aligned (a gene is upregulated in $S_x$ and $S_y$) or conflicting (a gene is upregulated in $S_x$ and downregulated in $S_y$), are still unclear.

To better understand the rules governing gene expression in combined stresses, we compared the gene expression changes of single stresses and combined stresses. For each of the seven stresses, represented by $S_x$, we identified significantly down- (blue), up-regulated (red), and unchanged genes, resulting in nine possible combinations of $S_x$ and $S_y$ (Fig. 6a). Then, for each combination, we calculated the average log2fc in $S_x$, $S_y$, and $S_{xy}$. The resulting plot revealed simple near-additive rules governing gene expression. For example, downregulated genes in cold stress ($S_x$ log2fc-2.3, Fig. 6a, top left corner), when combined with downregulated genes in all other stresses ($S_y$ log2fc-2.5), resulted in more negatively downregulated genes in the combined stresses ($S_{xy}$ log2fc-2.9). This near-additive pattern was seen for all seven stresses for downregulated (first row) and upregulated genes (last row, Fig. 6a). Interestingly, when $S_x$, $S_y$ regulation types were conflicting (e.g., $S_x$ was up- and $S_y$ was downregulated), $S_{xy}$ showed log2fc values between the two single stresses, as would be expected from additivity (Fig. 6a).

The near-additive pattern is seen when all possible combinations of $S_x$ and $S_y$ are color-coded by the $S_{xy}$ outcome (Fig. 6b). For example, genes upregulated in $S_x$ and $S_y$ tend to be more upregulated

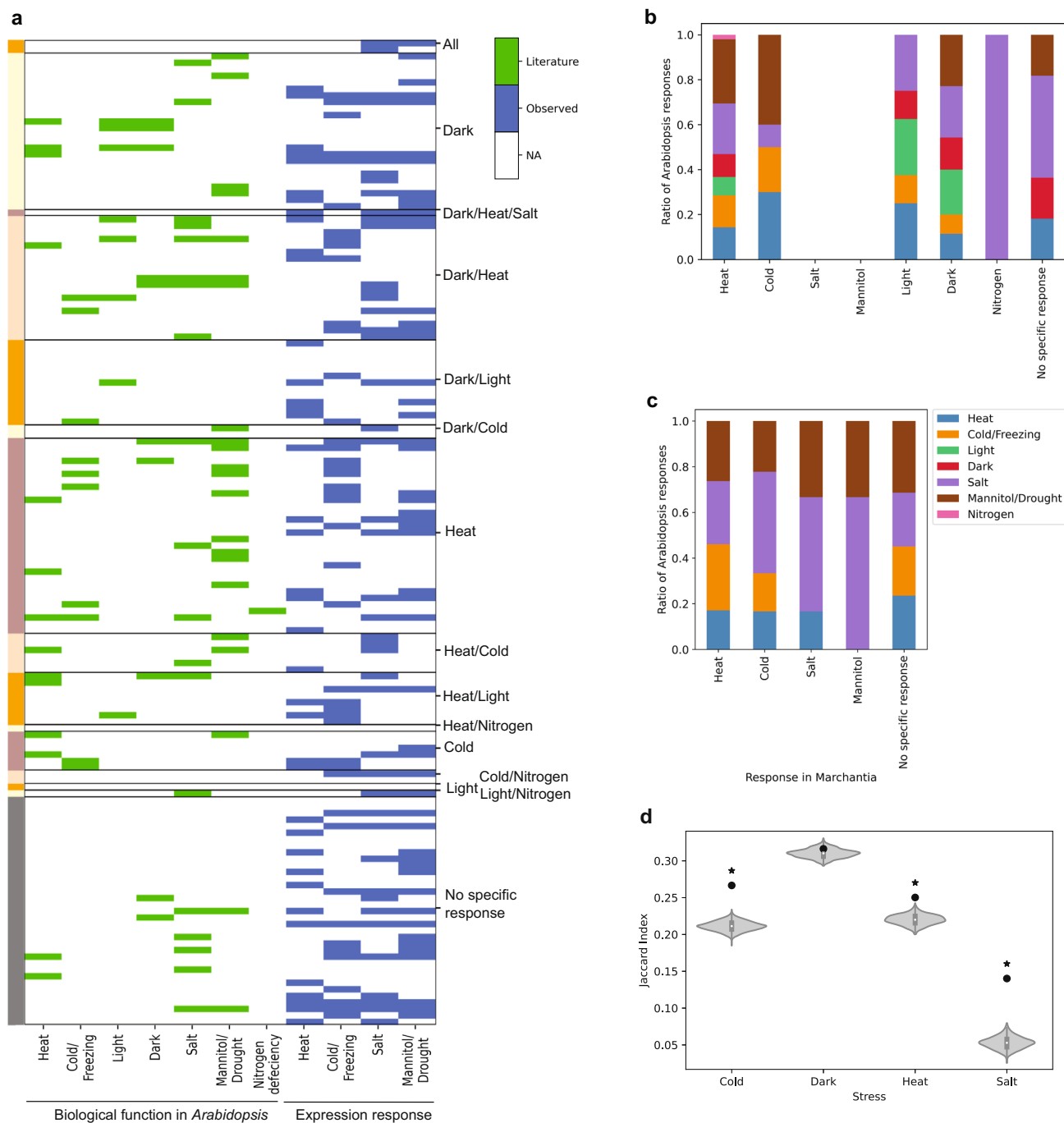

**Fig. 5 | Comparison of stress-responsive transcription factors in *Marchantia* to *Arabidopsis* orthologs. a** The function of *Arabidopsis thaliana* orthologs, inferred from the literature (green, obtained from NCBI, Arabidopsis.org) and gene expression responses (blue, eFP browser). Each row contains one *Arabidopsis* TF, and the rows are grouped and color-coded according to the stress response in *Marchantia*. The columns indicate the stresses observed in *Arabidopsis*. **b** Ratio of evidence from the literature for *Arabidopsis* TFs grouped according to the stress specificity observed in *Marchantia* (*x*-axis). **c** Ratio of evidence by observation of change in expression in *Arabidopsis* TFs based on expression data (source eFP browser). High light, dark, and nitrogen deficiency are omitted due to the lack of data. 'No specific response' comprises transcription factors that were not robustly

responding to a stress in *Marchantia*. **d** Similarity in differential gene expression between *Marchantia* and *Arabidopsis* homologs belonging to the same gene family. The black points indicate the observed JI capturing the similarity of the differentially expressed gene families. The violin plots indicate the JI where the gene-gene family assignments have been shuffled 1000 times. The black asterisk shows cases where the observed JI is significantly higher (*p*-value < 0.05). The center of the boxplot indicates the median. The upper and lower bounds of the box indicate 75th and 25th percentile, respectively. The whiskers indicate the 1.5x interquartile range, while the minima and maxima indicate the minimum and maximum Jaccard Index. The *p*-values were calculated using hypergeometric test with 1000 permutations.

in $S_{xy}$ (Fig. 6b, red points in upper right quadrant). Conversely, downregulation in $S_x$ and $S_y$ produces an even stronger downregulation in $S_{xy}$ (Fig. 6b, blue points in the lower left quadrant). Conversely, the conflicting $\log_2 fc$ values tend to produce a response

between $S_x$ and $S_y$ (Fig. 6b, gray points). Differential gene expression analysis of $S_x$, $S_y$, and $S_{xy}$ follow similar patterns, where downregulated $S_x$ and $S_y$ almost exclusively result in downregulated $S_{xy}$ (Fig. 6c, top row).

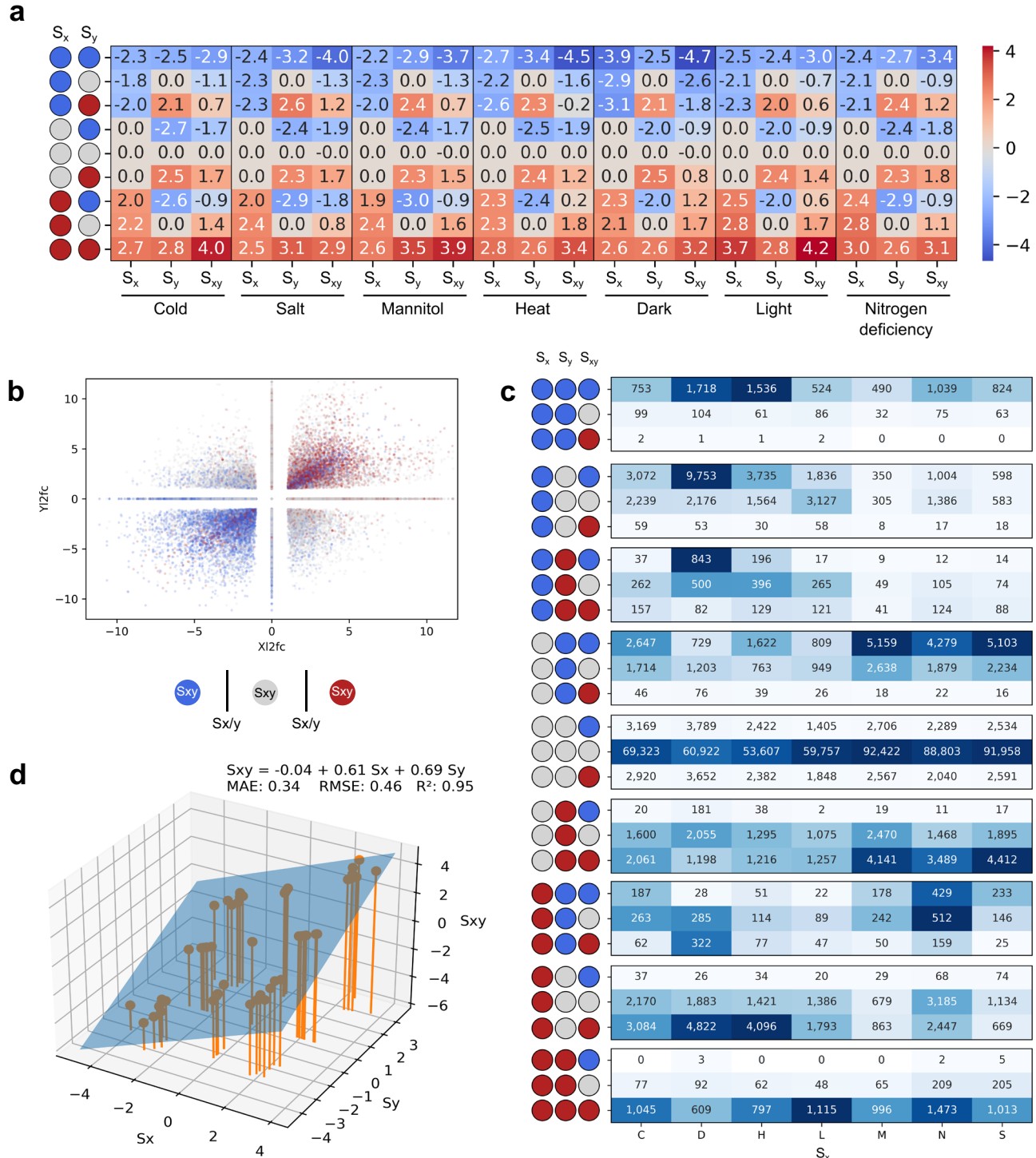

**Fig. 6 | Analysis of gene expression responses in combined stress. a** Averaged log$_2$fc of genes in S$_x$ (type of stress indicated at the bottom), S$_y$ (other stresses), and S$_{xy}$ (combined stress). Blue, red, and gray circles indicate genes that are significantly downregulated, upregulated, and not changed, respectively. **b** Scatter plot depicting the outcome in combined stress, where red, blue, and gray dots indicate that the response is higher, lower, or within the range of the single stresses, respectively. **c** Break down of the response observed in combined stress S$_{xy}$. The heatmap reflects the proportion of events in a given stress (column), and the colors are normalized across each category of S$_x$ and S$_y$. The actual number of observations is annotated in the cells. Blue, red, and gray circles indicate genes that are significantly downregulated, upregulated, and not changed, respectively. **d** Linear regression of the average log2fc values from panel (**a**). The inferred formula is shown, together with mean absolute error (MAE), root mean squared error (RMSE), and $R^2$ (goodness-of-fit measure).

Since the rules of how the different stress responses seem to follow a simple additive pattern, we investigated whether we can explain the average log$_2$fc S$_{xy}$ observed in Fig. 6a by regressing it on the average log$_2$fc S$_x$ and S$_y$. We found that the model S$_{xy}$ = −0.04 + 0.61*S$_x$ + 0.69*S$_y$ can excellently explain the average log$_2$fc ($R^2$ = 0.95, Fig. 6d), suggesting

a simple linear mechanism of integrating gene expression changes. To further examine how well S$_{xy}$ can be explained by the log$_2$fc values of the individual genes, rather than averages (Fig. 6d) from the perspective of various stresses, we performed another 3-dimensional linear regression (Fig. 7a–g), where the *x*-axis (S$_x$) and *y*-axis (S$_y$) of the cold-

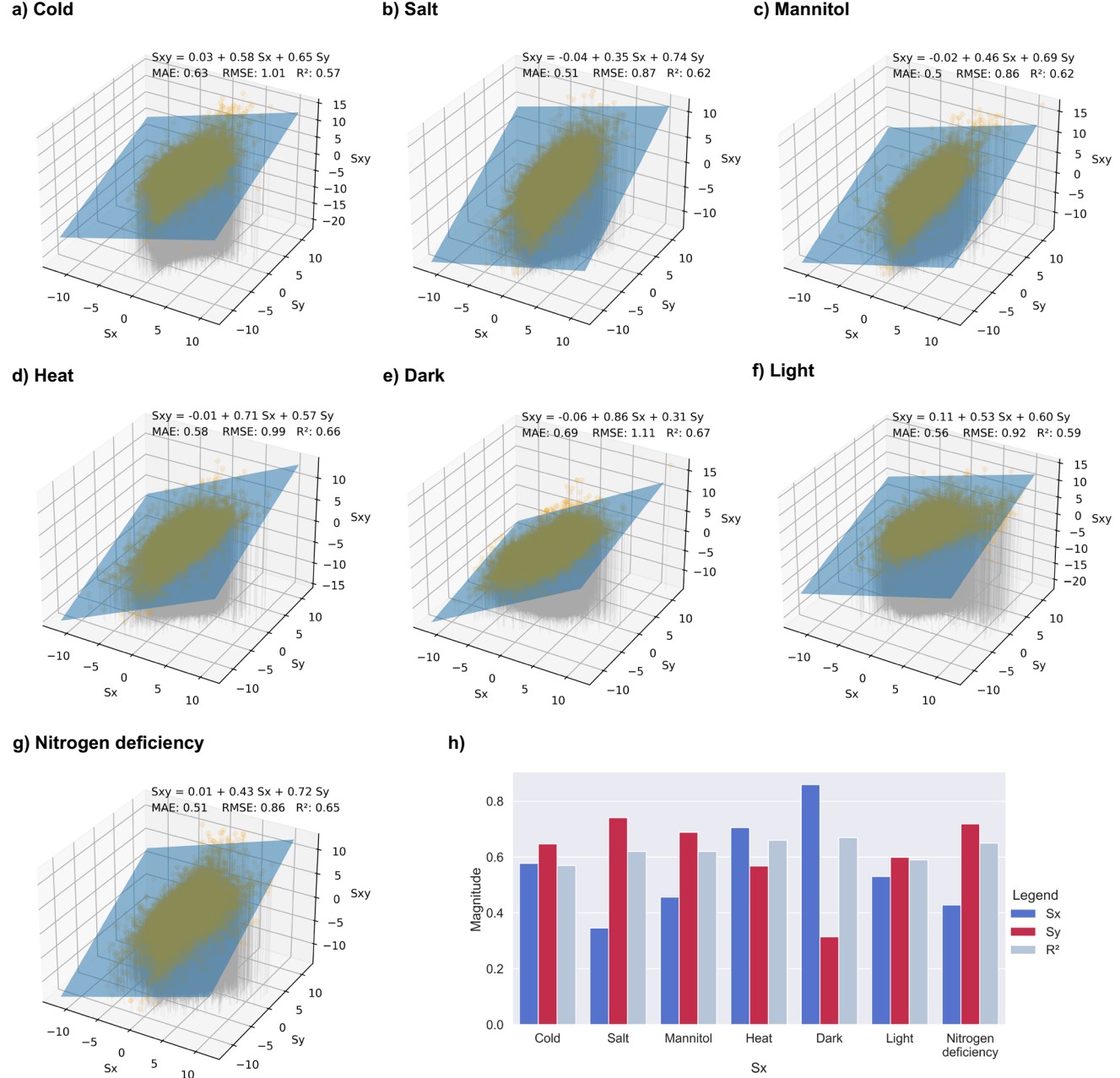

**Fig. 7 | Stress-specific linear regression analysis.** $S_x$ indicates log$_2$fc from the specific stress, $S_y$ indicates log$_2$fc from all other stresses, and $S_{xy}$ represents the log$_2$fc from the combined stresses. The $S_x$ stresses are **a** Cold, **b** Salt, **c** Mannitol, **d** Heat, **e** Dark, **f** Light, and **g** Nitrogen deficiency. **h** Summary of the linear regression coefficients and model prediction quality. The blue and red bars indicate the $S_x$ and $S_y$ coefficients, respectively, while the light blue bar indicates the $R^2$ value.

centric (Fig. 7a) plot represent individual gene expression values from cold stress and all other stresses respectively. While the $R^2$ values dropped, which is not unexpected as the first analysis in Fig. 6d was done on averaged log$_2$fc values which likely smoothed out gene-specific variations in gene expression, the resulting models could still approximate the expected values well ($R^2 = 0.57$–$0.67$).

Interestingly, we observed that the $S_x$ of $S_y$ parameters differed between stresses (Fig. 7h). For example, the $S_x$ parameter (reflecting log$_2$fc from the darkness experiment) in darkness is larger (0.86) than the $S_y$ parameter (0.31, capturing log$_2$fc from all non-darkness experiments), indicating that gene expression differences resulting from darkness have a higher influence on gene expression than other stresses, which is in line with above results (Fig. 2g). Based on the $S_x$ coefficients, we can rank the dominance of stresses as darkness (0.86) > heat

(0.71) > cold (0.58) > light (0.53) > mannitol (0.46) > nitrogen deficiency (0.43) > salt (0.35).

### eFP browser and CoNekT database for *Marchantia*

Bioinformatic data is only as useful as its accessibility. To make our data readily accessible, we have constructed an eFP browser instance for *Marchantia* available at https://bar.utoronto.ca/efp_marchantia/cgi-bin/efpWeb.cgi[33], and updated the CoNekT database[28] with our stress data, available at https://conekt.sbs.ntu.edu.sg/. To exemplify how our data and these databases can be used, we provide an example with phenylpropanoids, which contribute to all aspects of plant responses towards biotic and abiotic stimuli. In flowering plants, phenylpropanoids were found to be highly responsive to light or mineral treatment, important for resistance to pests[35], and invasion of

new habitats and reproduction[36]. The biosynthesis of phenylpropanoids begins with phenylalanine ammonia lyases (*PAL*) and tyrosine ammonia lyases (*TAL*) that catalyze the non-oxidative deamination of phenylalanine to trans-cinnamate, which directs the output from the shikimate pathway phenylpropanoid metabolism[35].

Notably, there are two other databases dedicated to *Marchantia*, MarpolBase[37] and MarpolBaseExpression (MBEX)[38], which are genome and expression databases respectively. With various information and functionality spread across databases, we summarize the main functionalities for the databases and the usefulness of each for different purposes. MarpolBase is the place to go for the download of genome-related information and tools such as BLAST to *Marchantia* and other plant species of interest, guide RNA design for CRISPR/Cas9, and other tools that aid in the retrieval of gene-related information. On the other hand, MBEX, eFP, and CoNekT are expression-based databases. Among the three, eFP serves purely as a visualization of expression data in the anatomy of the plant, and the *Marchantia* instance is one of the many species found in the database collection. Visualization by anatomy is also available on MBEX as 'Chromatic Expression Images'. While MBEX and CoNekT overlap in the area of expression and co-expression tools, MBEX provides more downstream analysis tools that allow for the analysis of differential expression, enrichment of biological functions, and set relations between conditions in *Marchantia*. On the other hand, CoNekT provides an organ-centric dissection of the expression data with the integration of orthology information across 12 other species from the Viridiplantae and tools for comparison across species and tissues. However, only CoNekT and eFP browser contain the single and combined stress data at the moment.

To study phenylpropanoid metabolism in *Marchantia*, we started by entering 'PAL1' into CoNekT's search box (https://conekt.sbs.ntu.edu.sg/), which took us to the page of *PAL1*, a *PAL* gene *AT2G37040* from *Arabidopsis thaliana*. To identify *PAL* genes in *Marchantia*, we clicked on the link of the Land Plants orthogroup (CoNekT provides orthogroups for Archaeplastida, Land Plants, and Seed Plants), which revealed that all 11 land plants in the database contain *PAL*s, while the algae *Cyanophora paradoxa* and *Chlamydomonas reinhardtii* do not. *Marchantia* contains a surprisingly large number (13) of *PAL*s when compared to *Arabidopsis* (4). CoNekT contains gene trees that also show the expression of genes in major organ types. The analysis revealed that the many *PAL* genes in *Marchantia* likely resulted from a recent duplication within Marchantia (Supplementary Fig. 16).

To gain insight into the function of the 13 *PAL* genes from Marchantia, we set out to study the expression of the *PAL*s during stress conditions. First, we copied the gene identifiers into Tools/Create heatmap (https://conekt.sbs.ntu.edu.sg/heatmap/), revealing that most *PAL*s show a high expression during cold treatment combined with nitrogen starvation (Fig. 8a). Then, we clicked on one of the highly responsive genes *Mp4g14110.1*, which took us to the page dedicated to the gene. The page contains various information, such as gene description, CDS/protein sequences, gene families, found protein domains, co-expression networks, and others. The detailed CoNekT expression profile plot (Fig. 8b) and eFP browser results (Fig. 8c, red square at the intersection of cold and nitrogen deficiency) confirmed the high cold + nitrogen-specific expression of *Mp4g14110*. To better understand the function of *Mp4g14110*, we clicked on the cluster link that directed us to the co-expression cluster link of *Mp4g14110*. The cluster page contains information about the genes found in the cluster, Gene Ontology enrichment analysis, found protein domains, and other functional information.

Interestingly, the 'Similar Clusters' section of the cluster page revealed similar co-expression clusters in other species (similarity is based on ortholog membership and defined by JI), and with another similar clusters in *Marchantia*. To study these duplicated clusters, we clicked on the 'Compare' button, which revealed that the two clusters contained several gene families involved in phenylpropanoid

biosynthesis (yellow rounded rectangles: *PAL*s, brown rectangles: chalcone synthases), *ABC* and *DTX* transporters implicated in metabolite transport across membranes (purple/gray/green and red rectangles[39]), auresidin synthases that can hydroxylate or cyclize chalcones (light blue rounded rectangles[40]) and glutathione transferases (red rounded square[41]) (Fig. 8d). Interestingly, both clusters contained WRKY TFs (salmon rectangles), implicating these TFs in controlling the biosynthesis of the respective metabolites.

Taken together, our tools revealed evidence of duplicated modules, likely involved in the biosynthesis of related phenylpropanoids. The updated CoNekT platform contains many additional tools to predict gene function and find relevant genes[16,28,34,42]. For example, the tool found many of the described genes by identifying genes highly expressed during combined cold and nitrogen starvation (Supplementary Data 10), by clicking on Tools/Search Specific Profiles, selecting *Marchantia* and 'Cold-Nitrogen deficiency'.

## Discussion

Plants are often exposed to multiple abiotic stresses, which require them to perceive and integrate multiple signals and respond in a manner that allow them to survive. To understand how plants integrate and respond to the various environmental cues, we constructed a stress expression atlas capturing gene expression changes to single and combined stresses for *Marchantia*.

Notably, the chronic stresses darkness, salt, osmotic, and nitrogen deficiency cause growth inhibition largely due to the longer duration of these stresses, likely capturing late-stage responses. During the comparison of chronic stresses with the acute stresses of heat, cold, and light, we may miss the effects of early-stage responses that might have been present in the acute stresses. However, we assume that the sampling of 1 day after stress induction for acute stresses captures mostly late-stage responses, allowing us to compare the chronic and acute stresses[43].

To determine the response of *Marchantia* towards various single stresses, we tested a range of severity for heat, cold, salt, osmotic, light, dark, and nitrogen deficiency on *Marchantia*. As expected in most stresses, the size of the plants decreased proportionally to the severity of stresses (Fig. 1b). Specific stresses, such as 3 °C cold and 535 uEm$^{-2}$s$^{-1}$ light, caused only minor growth phenotypes, suggesting that *Marchantia* can survive under even lower temperatures and higher light intensities. Conversely, stresses such as osmotic (100 mM mannitol), salt (40 mM NaCl), and carbon starvation (3 days of darkness) produce strong growth phenotypes resulting in reduced growth and discoloration (yellow-green for mannitol >150 mM, darker plants for salt >60 mM salt) (Fig. 1b). Death of plants occurred for heat stress at 36 °C for 24 h and at higher salt concentrations (>200 mM). Interestingly, the plants grew at 0 mM KNO$_3$, albeit slower and with reddish discoloration likely caused by auronidin, a flavonoid shown to accumulate during phosphate deficiency[29].

In most cases, the combination of two stresses resulted in additive phenotypes (e.g., salt+mannitol stress results in smaller plants than the two stresses separately, Fig. 1c−e). While heat (33 °C) combined with high light intensity (435 uEm$^{-2}$s$^{-1}$) resulted in death, this was caused by temperature built up in the sealed plates, causing the temperature to rise to the lethal 38 °C. Consequently, this stress combination should be performed using an open plate setup in the future. The only exception to this was salt+nitrogen deficiency (SN) (40 mM NaCl, 0 mM KNO$_3$), which was significantly larger than plants exposed only to salt stress and was observed to have extensive and dense rhizoids (Source Data 1). Curiously, we did not observe any dramatic phenotypes when combining carbon/energy starvation (3 days of darkness) with any other stresses (Fig. 1c). This is counterintuitive as, e.g. heat stress acclimation is a costly process requiring the biosynthesis of new transcripts and proteins to repair, replace and rebalance the affected cellular machinery[44]. A hypothesis for the unexpectedly mild

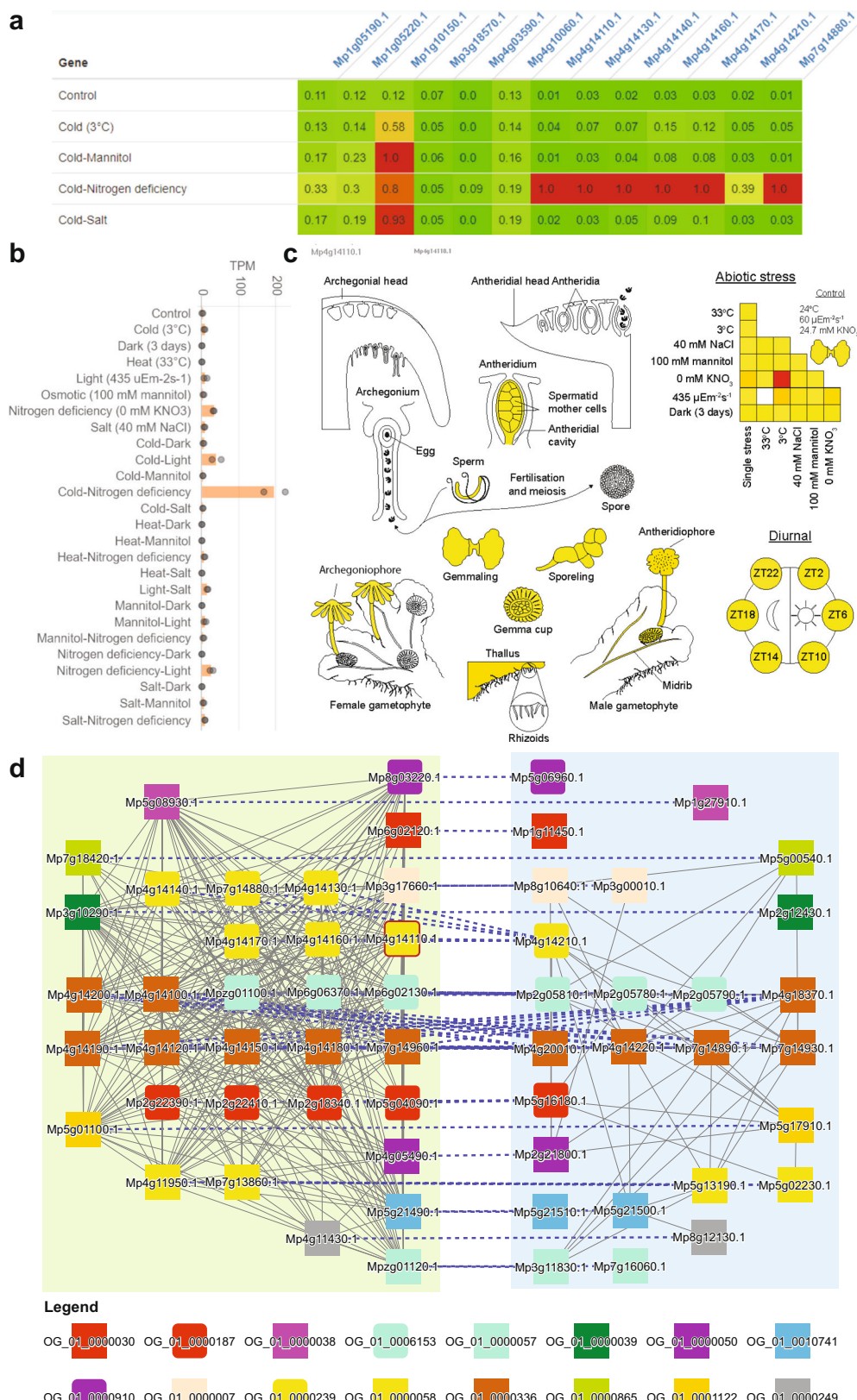

**Fig. 8 | Implementation of the Marchantia gene expression data in CoNekT and eFP browser. a** Heatmap showing expression of *PAL*s in representative stresses. **b** Expression profile of *Mp4g14110.1* (Mp*PAL7*) under single and combined stress in CoNekt. **c** Expression of Mp*PAL7* in different organs and under different abiotic stress conditions in eFP. **d** Comparison of Cluster 102 and 50 bounded by green and blue boxes, respectively. The border of node Mp*PAL7* is colored red, and blue dashed lines indicate homology between genes from the two clusters.

phenotype for combined dark stress could be due to the lack of photosynthetic processes in the absence of light, which increases the plants' capacity to cope with increased ROS, a typical response to most abiotic stresses[45]. This is also exemplified by the significant downregulation of oxidoreductases and chloroplast redox homeostasis in combined dark stress (Fig. 2g). These phenotypes demonstrate that *Marchantia* is able to survive various adverse growth conditions and can serve as an excellent model for studying stress acclimation.

Interestingly, environmental stresses (cold, heat, darkness, and high light) showed a higher number of DEGs than media stresses (salt, mannitol, and nitrogen deficiency) (Fig. 2a). We speculate that the environmental stresses resulted in more DEGs because these stresses can permeate every cell, affect every protein (heat, darkness), and/or dramatically affect the energy levels that have consequences on all processes (darkness, high light). Conversely, stresses such as mannitol and salt can be effectively contained by the action of ion transporters and osmolyte accumulation[46]. Interestingly, we did not observe any correlation between the number of DEGs and the effect on plant growth (Supplementary Fig. 4B); while salt and mannitol treatments caused the most dramatic growth defects, these two stresses also showed the lowest number of DEGs (Supplementary Fig. 4B). This observation is in line with our study on alga *Cyanophora*[16], suggesting that there is no correlation between a visible phenotype and growth across stresses in other plants.

We observed that certain stresses dominate the transcriptional responses. For example, darkness+cold looks more like darkness than cold (Fig. 2g), and our metric measuring suppression suggests that darkness is the dominant stress (Fig. 2e, f, plots are red in darkness stresses). The analysis of differentially expressed pathways allowed us to rank the strength of dominance of abiotic stresses: darkness (clustered in six stress combinations), then heat and light (four combinations each), then nitrogen deficiency and cold (three each) (Fig. 2g). To better understand the mechanism governing the dominance of the stresses, we performed several analyses that revealed multiple mechanisms that likely work together.

Firstly, we identified 75 robustly-responding TFs, revealing that the dominant stresses (e.g., heat and darkness) differentially express a higher number of TFs than the non-dominant stresses (Fig. 3c). Secondly, the inferred TFRN showed that the TFs active in the dominant stresses regulate more genes and other TFs than TFs from non-dominant stresses (Fig. 4a nodes labelled with H and D tend to be darker). Thirdly, our regression model showed that the gene expression changes in a combination of two stresses could be explained by the addition of the $\log_2fc$ values (Fig. 6d). Thus, when combining stress with a high number of largely negative $\log_2fc$ values (e.g., darkness downregulated $\log_2fc$ is between −3.9 to −2.9, Fig. 6a, top row), with other stresses, the $\log_2fc$ in combined stress will also be negative ($\log_2fc$ for combined stress in darkness down-regulated genes ranges between −4.7 to −1.8, Fig. 6a). Fourth, our regression models showed that dominant stresses have higher coefficients (Fig. 7h), suggesting yet another unknown component governing the integration of multiple stress responses.

Our comparison of stress-responsive *Marchantia* TFs and their *Arabidopsis* orthologs revealed that majority of *Arabidopsis* TFs were not involved in the same stresses as the *Marchantia* TFs (Fig. 5a, c). On the other hand, we did observe significant conservation in differential gene expression between *Arabidopsis* and *Marchantia* gene families in cold, heat, and salt, but not dark (Fig. 5d). The latter result agrees with studies showing the conservation of stress response in plants focusing on transcription factors and hormones[47–51]. However, the relatively poor conservation between *Marchantia* and *Arabidopsis* orthologs is not entirely unexpected, as massive changes such as genome and gene duplications have occurred since the last common ancestor, rendering a lack of homology in genes across species and differences in gene families and regulation. This suggests that each model plant uses a

different strategy to acclimate to stress, and considerations regarding the phylogenetic distance between model species and crops of interest should be accounted for when attempting to make knowledge transfers from a model plant to crops. This lack of conservation of responses to abiotic stresses has been observed by us at the species level in *Cyanophora*[16], and at the intraspecies level by a salt stress study in six Lotus accessions, where only 1% of genes showed a conserved response[52], in seven *Arabidopsis* accessions, which showed a divergent response to treatment by salicylic acid[53] and by two strawberry cultivars, which displayed modest conservation of DEGs to the same pathogen[54]. However, while we analyzed only gene expression, stress responses can be active at the epigenetic (methylation of genes), transcriptomic (mRNA, microRNA, lncRNA), and proteomic (post-translational modifications and activation) levels[55–59].

To make our data more readily accessible, we provide an eFP browser for *Marchantia*, a popular tool allowing the visualization of gene expression by an 'electronic Fluorescent Pictograph'. Furthermore, we provide an updated CoNekT database with expression atlases of 13 species comprising various algae and land plants. The database provides tools to study gene expression, functional enrichment analyses of co-expression networks, and other comparative tools. These valuable tools will help further dissect the gene regulatory networks behind abiotic stress responses in *Marchantia* and other species (Fig. 8).

Importantly, our analysis shows that it is possible to predict gene expression of combined stresses with a simple linear regression. This paves the way to building more complex, better-performing models that can predict gene expression in any environment, given sufficient input data. This strengthens the call for more emphasis on studying combined biotic and abiotic stresses in light of future challenges posed by climate change[60,61].

## Methods

### Maintenance of *Marchantia*
Male *Marchantia polymorpha*, accession Takaragaike-1 was propagated on half-strength Gamborg B-5 Basal agar (1% sucrose, pH 5.5, 1.4% agar)[62] in deep well plates (SPL Biosciences, SPL 310101) at 24 °C under continuous LED light at 60 µEm$^{-2}$s$^{-1}$.

### Experimental setup for stress experiment
To determine the ideal stress condition for cross-stress experiments, a series of severity levels was used for heat, cold, salt, osmotic, light, dark, and nitrogen deficiency stresses. For each stress level, three agar plates containing nine gemmae each were plated, where two plates were used as material for RNA sequencing, and one plate was kept for observation until 21 days after plating (DAP).

For salt and osmotic stress, the gemmae were plated on half-strength Gamborg B-5 Basal agar (pH 5.5, 1.4% agar, 12.4 mM KNO$_3$, 0.5 mM (NH$_4$)$_2$SO$_4$ supplemented with 20–200 mM NaCl (20 mM steps), and 50–400 mM mannitol (50 mM steps) respectively. For nitrogen deficiency stress, the gemmae were plated on half-strength Gamborg B-5 Basal agar (pH 5.5, 1.4% agar, 0.5 mM (NH$_4$)$_2$SO$_4$) and KNO$_3$ concentration ranging from 90 to 0% (12.4–0 mM, 1.2 mM steps), respectively. The potassium concentration in nitrogen deficiency agar was maintained using equimolar concentrations of KCl. Gemmae subjected to heat, cold, light, and dark stress were plated on normal half-strength Gamborg B5 agar (pH 5.5, 1.4% agar, 12.4 mM KNO$_3$, 0.5 mM (NH$_4$)$_2$SO$_4$).

Plates were grown at 24 °C under continuous LED light at 60 µEm$^{-2}$s$^{-1}$ from days 0 to 13. For dark stress, plates were moved to the plant growth chamber (HiPoint M-313) on days 8, 9, 10, 11, 12, 13, and 14 for growth in darkness at 24 °C for 7, 6, 5, 4, 3, 2 and 1 day(s), respectively. On day 14, all plates were transferred to the plant growth chamber (HiPoint M-313) for 24 h. Control and plates for salt, osmotic, and nitrogen deficiency stresses were maintained at 24 °C

under continuous LED light at 60 µEm$^{-2}$s$^{-1}$ in the plant growth chamber. The following modifications were used growth conditions for heat, cold, dark, and light stresses. For heat and cold stress, 27 °C to 36 °C (3 °C steps) and 3 °C to 12 °C (3 °C steps) were used, respectively. For light stress, plants were subjected to 20 to 100% chamber capacity of light output (115 to 535 µEm$^{-2}$s$^{-1}$) at steps of 20%, ~100 µEm$^{-2}$s$^{-1}$ per step.

On day 15, whole plants were harvested by pooling three plants into 2 mL Eppendorf tubes, flash-frozen in liquid nitrogen, and stored at −80 °C. This was done in triplicates, resulting in 3 (replicates) × 3 (pooled plants) used for each RNA-seq sample. Images of the front and back of the observation plates were taken, and the plates were returned to normal growth conditions. Pictures of the front and back of the observation plates were taken on day 21 without cover. All pictures are collated in Source Data 1.

Cross-stress experiments were carried out similarly with the following parameters for the various stress combinations - heat (33 °C), cold (3 °C), salt (40 mM NaCl), osmotic (100 mM mannitol), light (435 µMm$^{-2}$s$^{-1}$), darkness (3 days), and nitrogen deficiency (0 mM KNO$_3$).

Due to equipment constraints, the experiment was carried out over 22 batches. Some variability in size was observed in the controls (Supplementary Fig. 1). To account for this variability, we have sequenced controls from batches F and L to impose a more stringent criteria for the identification of differential gene expression.

### Size measurement of *Marchantia*
To measure the size of the plants grown under the different stresses, the images of the observation plates were scaled and measured in Adobe Illustrator. The length and breadth of the thallus were taken in relation to the central axis (Fig. 1b, c show the plants with the central axis in a vertical position). The thallus's approximate area was derived from the product of the length and breadth of the thallus. Abnormally small plants (outliers) were excluded from further analysis.

### RNA extraction and sequencing
Using a mortar and pestle, plants from each Eppendorf tube were ground into fine powder in liquid nitrogen. Total RNA was extracted using the Spectrum™ Plant Total RNA Kit (Sigma, STRN-250) using Protocol A (750 µL Binding Solution) according to manufacturer's instruction with on-column DNase digestion using 60 µL of DNase mixture (15 µL RQ1 RNase-Free DNase (Promega, M6101), 6 µL RQ1 DNase 10X Reaction Buffer and 19 µL nuclease-free water) per column.

Preliminary quality control of the extracted RNA (triplicates for each condition) was done using Nanodrop before further quality control checks by Novogene (Singapore) for sample quantitation, integrity, and purity using Nanodrop, agarose gel electrophoresis and Agilent 2100 Bioanalyzer. Library construction from total RNA, including eukaryotic mRNA enrichment by oligo(dT) beads, library size selection, and PCR enrichment, was performed by Novogene using NEBNext® Ultra™ II Directional RNA Library Prep Kit for Illumina®. The libraries were then sequenced with Illumina Novaseq-6000, paired-end sequencing at 150 base pairs, and sequencing depths of ~20 million reads per sample.

### Expression quantification
RNA sequencing data were mapped against the *Marchantia poly-morpha* CDS (v5.1 revision 1, MarpolBase[37]), quantified, and TPM-normalized (transcript per million) using Kallisto v 0.46.1[63].

### Identification of differentially expressed genes
Non-normalized counts from Kallisto were used to analyze differentially expressed genes (DEGs) using R package DESeq2[64], where various

stress conditions were compared against the controls from batches F and L.

For our *Marchantia* dataset, only genes that were found to be differentially expressed against controls from two different batches, F and L, were considered for further analysis. For downstream analysis, only genes with a Benjamini−Hochberg[65] adjusted $p$-value <0.05 and a−1> log$_2$fc >1 were considered as differentially expressed.

### Identification of transcription factors and differentially expressed pathways
The biological function and pathway membership of genes of *Marchantia* were annotated using Mercator 4 v2.0[66]. The annotation of *Marchantia* TFs was retrieved from PlantTFDB v5.0[67].

Significantly differentially expressed pathways were determined through a permutation analysis, where the observed number of DEGs in a pathway was compared to the permuted number of DEGs. The $p$-values were adjusted for multiple testing using Benjamini−Hochberg correction ($p$-value < 0.05)[65]. To identify which stresses (columns) and biological pathways (rows) were similar, we first calculated the Jaccard distance (JD) between all columns/rows. The JD values were used to build a condensed distance matrix which was used as input for the hierarchical clustering algorithm.

To investigate the behavior of homologous TFs in *Arabidopsis* and *Marchantia* during various stress conditions, we compared the occurrence of DEGs from our dataset and from Ferrari et al.[16] study on *Arabidopsis* cold (E-GEOD-63406), dark (E-GEOD-67956), heat (E-GEOD-72806), and salt stresses (E-GEOD-72806). Corrected $p$-value (<0.05) and an absolute log$_2$fc cut-off of more than 1 were used to determine DEGs for both datasets. Homologous relationships between *Arabidopsis* and *Marchantia* were obtained from PLAZA Dicots 5.0[68] and the compatibility of gene names from *Marchantia* genome v3 to v5.1r1 was ensured using the conversion table on MarpolBase[37]. For the analysis, only homolog IDs common to both species were used. The JI based on the homolog ID assigned to the gene in PLAZA was calculated for each stress. In addition, the $p$-value was determined through a permutation test where the observed JI of homolog IDs in a stress for each species was compared to the permuted JI of homolog IDs 1000 times.

### Construction of stress-specific gene regulatory networks
To reconstruct the gene regulatory network (GRN), we selected DEGs expressed in more than five experiments to ensure sufficient variability in our dataset needed for statistical modeling (Supplementary Fig. 8A). Apart from using all the experiments, we also reconstructed stress-specific networks by employing a subset of experiments that included the respective stresses (Supplementary Fig. 8B). For example, the heat-specific network is based only on data from experiments where heat stress was involved (i.e., Heat, Heat-Mannitol, Heat-Salt, Heat-Dark, and Heat-Nitrogen deficiency).

The gene expression values were log-transformed and scaled prior to modeling; hence, the relationship between the expression of a DEG, as a response variable, and the expression of TFs, as predictor variables, is non-linear. The GRN was reconstructed using linear regression as described in Eq. (1) where the response variable, $Y$, represents the vector of (log-transformed) expression levels of a DEG; the predictor variables, $X$, represents a matrix of (log-transformed) expression levels of the TFs; and the error term, $\varepsilon$, of normally distributed variance. The parameters (i.e regression coefficients), $\beta$, were estimated by minimizing the sum of the loss and a weighted combination of the first and second norm of the regression coefficients, termed ElasticNet regularization[69] as described in Eq. (2), which provided a good compromise between model sparsity (i.e., feature selection corresponding to the inclusion of transcription factors in the

model) and model explanatory power.

$$Y = X\beta + \varepsilon \tag{1}$$

$$L_{enet}(\hat{\beta}) = \frac{\sum_{i-1}^{n}(y_i - x_i'\hat{\beta})^2}{2n} + \lambda\left(\frac{1-\alpha}{2}\sum_{j-i}^{m}\hat{\beta}_j^2 + \alpha\sum_{j-1}^{m}|\hat{\beta}_j|\right) \tag{2}$$

Three- and five-fold cross-validation were used for the stress-specific data and all data, respectively, to determine the optimal $\lambda$ (L1 regularization; 0.1 to 0.9, 0.1 steps) and $\alpha$ (L2 regularization; 0, 0.001, 0.01, 0.05, 0.1, 0.5, 1, 1.5, 2, 10, 100) that resulted in the lowest coefficient of variation for each model. We filtered for good quality models with $R^2 > 0.8$ (Supplementary Fig. 9A).

### Evaluating the accuracy of the GRNs

We evaluated our GRN network against the known curated gene regulatory networks in *Arabidopsis* retrieved from AGRIS[70]. Orthogroups of *Arabidopsis* and *Marchantia* genes were identified using Orthofinder v2.3.1[71] and used as the basis for comparing the *Arabidopsis* and *Marchantia* GRNs. We used the JI to quantify the similarity between the GRNs, where TF-target edges were converted into orthogroup-orthogroup tuples that were used in the set comparisons. The union of the stress-specific networks produced the highest JI score between our networks and the AGRIS network (Supplementary Fig. 9B). To test the significance of the similarity between our GRN and the AGRIS network, we calculated empirical p-values by shuffling the TF and gene pair in the AGRIS network 1000 times and calculated the resulting JI for each shuffling.

### Construction of the high-confidence GRN

For each gene, we identified the TF with the highest absolute relative coefficient (Fig. 3a) in the merged network. In the merged network, the coefficients in different stress-specific networks are used to determine whether the TF is an activator (all coefficients are positive), repressor (all coefficients are negative) or if the regulation is ambiguous (mixture of positive and negative coefficients). For example, if TF *X* regulates gene *Y* in 4 different stress-specific networks with a positive coefficient, the TF is considered an activator.

### Revealing transcription factors regulating biological pathways

To understand how robustly-regulated TFs might be affecting certain biological processes, we took TFs and second-level Mapman bins[66] that were specifically regulated in a stress group. A stress group is defined as a group of experiments sharing common stress, for example, the heat stress group contains the experiments Heat, Heat-Mannitol, Heat-Salt, Heat-Dark, and Heat-Nitrogen deficiency.

Robustly-responding TFs were identified based on the ratio of occurrences where it is differentially regulated in a stress group against the number of experiments in the stress group. A TF is considered to be specifically expressed in the stress group if the ratio >0.7. Stress group-specific MapMan bins were defined in the same manner.

Robustly-responding MapMan bins (Supplementary Fig. 13) were considered to be regulated by robustly responding TFs if at least 5% of the genes (Supplementary Fig. 14) in the MapMan bin were regulated by the TF in our GRN (Figs. 4c, d and S15).

### Inference of TFRN

We chose the highest absolute relative coefficient for each TF-TF pair to indicate the putative regulatory relationships between TFs. Next, we defined the transcription factors to be upregulated, downregulated, or ambiguous (up- and down-regulated in more than 1 stress group) based on their specific expression across stress groups. We then defined the edges as expected if: (1) $TF_A$ (up-/down-regulated, activator) regulates $TF_B$ (up-/down-regulated), (2) $TF_A$ (down-/up-regulated, repressor) regulates $TF_B$ (up-/down-regulated). All other edges were defined as unexpected, e.g., $TF_A$ (upregulated, activator) regulates $TF_B$ (downregulated). Finally, we applied an absolute coefficient cut-off that produced the highest ratio of expected / total edges (Supplementary Fig. 11), arriving at a cut-off value of 0.22.

### Linear regression of gene expression

Trends in gene expression during combined stress were revealed using ordinary least squares (OLS) regression. For the general trend, we first grouped the expression values ($\log_2 fc$) according to the 9 possible combinations of responses in Sx and Sy (i.e. downregulated/upregulated/no change in Sx and Sy) for each stress, where Sx represents the stress of interest. The $\log_2 fc$ values were then averaged. In addition, linear regression shown in Fig. 7a–g was performed on non-averaged gene expression values for each stress group independently. For example, for cold stress, this will involve Sxy of CD, CL, CN, CS, and CM, where Sx is C and Sy are all other corresponding stresses. A standard linear regression of the (averaged) $\log_2 fc$ was performed using the scikit-learn linear model package (linear_model) with Sxy as the response variable and Sx and Sy as the predictor variables.

### Functional analysis of *Arabidopsis* TF orthologs

The biological function of *Arabidopsis* TF orthologs was inferred from gene ontology terms with experimental evidence and literature searches. The expression responses were inferred through observation of gene expression changes on the *Arabidopsis* eFP browser using the "Abiotic stress" dataset from[33].

### Reporting summary

Further information on research design is available in the Nature Portfolio Reporting Summary linked to this article.

## Data availability

The RNA-seq data capturing the expression of controls, single and double stresses are available from https://www.ebi.ac.uk/ena as E-MTAB-11141 [https://www.ebi.ac.uk/biostudies/arrayexpress/studies/E-MTAB-11141]. Source data are provided with this paper.

## Code availability

Python and bash scripts used to generate the figures in the paper are available from: https://github.com/tqiaowen/marchantia-stress.

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

## Acknowledgements

We want to thank the Nanyang Technological University start-up grant and Singapore Food Agency grant SFS_RND_SUFP_001_05 and MoE Tier 2 No – 022580-00001 grant for funding. We want to thank the members of the Mutwil lab for their comments on the manuscript. We thank D. Maizels (http://www.scientific-art.com/) for the illustrations used for the eFP browser.

## Author contributions

M.M. conceived the project, wrote the manuscript, Q.W.T. performed the single stress experiments and data analysis, updated the CoNekT database and wrote the manuscript, P.K.L performed the combined stress experiments, N.P. and A.P. provided the eFP browser, Z.C., provided the *Marchantia* cultures and feedback on growing *Marchantia*. M.A. provided scripts for the ElasticNet regression, while Z.N. supervised M.A. and Q.W.T. in ElasticNet regression. Z.N. also contributed to manuscript writing and project design.

## Competing interests

The authors declare no competing interests.
