## [Peer Review File · Nature Communications]

Cross-stress gene expression atlas of *Marchantia polymorpha* reveals the hierarchy and regulatory principles of abiotic stress responsesReviewer #1 (Remarks to the Author):

SUMMARY:

The authors analyze data obtained from exposing the plant, *Marchantia polymorpha*, to seven individual stresses (heat, cold, salt, osmotic, light, dark, and nitrogen deficiency) and to 19 pairwise combinations of those stresses. The constructed an abiotic gene expression atlas of *Marchantia* to understand how plant integrate and respond to the various environmental cues. Genes that were found to be differentially expressed against controls from two different batches were considered for further analysis. DEGs expressed in 5 or more experiments were used to calculate gene regulatory networks (GRN) for all experiments and stress-specific networks using a subset of experiments that included the respective experiments. GRN were generated using linear regression with elastic net regularization. The authors identified 75 robustly-responding TFs, revealing that dominant stresses (e.g. heat and darkness) differentially express a higher number of TFs than the non-dominant stresses. The inferred GRNs showed that the TFs active in the dominant stresses regulate more genes and other TFs than TFs from non-dominant stresses. These regression GRNs also showed that gene expression changes observed from pairwise experiments could be explained by a linear combination of the log₂ fold change values from the single experiments. To make the resulting data more accessible, they provide an eFP browser for *Marchantia*.

REVIEW:

What are the noteworthy results?

The primary noteworthy result from the manuscript is the rich dataset capturing the transcriptional response for single and combinatorial stresses. The process for identifying relevant specifications for the single and combinatorial stresses experiments were systematic and produced a rich dataset that would be of interest to other researchers in the field.

The result identifying that combinatorial stresses can be described as linear combinations of the results from single stresses is particularly noteworthy. This tracks with the low level of redundancy seen in *Marchantia*. Also, the identification of a hierarchy of stress responses is also noteworthy.

Major Concerns:

- N/A

Minor Concerns:

- N/A

Will the work be of significance to the field and related fields? How does it compare to the established literature? If the work is not original, please provide relevant references.

The resulting data generated, along with the web interface, would be of significance to researchers in the field. This is a rather expansive data set that could be used for further analysis. The results here could also be further compared to combinatorial stress response results from other plants such as maize and *Arabidopsis* to better understand how these plants have evolved to handle stresses typically seen in nature.

Major Concerns:

- N/A

Minor Concerns:

- N/A

Does the work support the conclusions and claims, or is additional evidence needed?

It seems like the work broadly supports the major conclusions and claims. Further description and

refinement of the analysis and methodology would help to strengthen the manuscript.

Major Concerns:

- N/A

Minor Concerns:

- N/A

Are there any flaws in the data analysis, interpretation and conclusions? Do these prohibit publication or require revision?

There are significant flaws in the interpretation of some of the results. Please see below for specific points.

Major Concerns:

- On lines 422 to 432, a more formal interpretation beyond the heatmap shown in Figure 2 F and G would be appropriate. For example, is it fair to say that the similarity metric ranged from 5% similarity to 25% similarity. A quantitative assessment and interpretation would be useful.

- Based on the explanation in the manuscript, it seems like the results in Figure S9A is quantify the genes and their corresponding equation that has an R^2 value above a certain amount. Thus, setting a filter of $R^2 > 0.8$ indicates the genes where there is some level of confidence in the prediction. This is a different interpretation than what is shown in the Figure, which refers to Number of models. The use of the term "Number of models" infers that the approach was ran N number of times and from these N runs, these are the resulting models (with each model being comprised of the equations for all DEGs) with R^2 values $>$ some value. The interpretation of these results should be revised.

- There are places in the manuscript where the explanation of the results are confusing and or does not flow in a way that is easily followed. This is a result of the use of terminology not common in the field (see above comment regarding the reference of each DEG equation as a model) or the extrapolation of results from several figures without a clear connection. This is a major flaw and should be clarified.

Minor Concerns:

- N/A

Is the methodology sound? Does the work meet the expected standards in your field?
See below.

Major Concerns:

- The overall methodology for modeling of the GRN could be improved. While using these results to get a broad understanding of the linear relationship between TFs and potential target genes are noteworthy, this approach does not capture the state-of-art in building gene regulatory networks for biological systems. While the development of new approaches for building GRNs is not the goal of the manuscript, further justification for the methodology used is warranted. The manuscript could also benefit from a description of the pros and cons associated with using such an approach.

Minor Concerns:

- N/A

Is there enough detail provided in the methods for the work to be reproduced?
Significant improvement in the description of the computational methods is needed. See below comments for detailed recommendations.

Major Concerns:

- Please further describe in the Materials and Methods section the construction of the linear regression with elastic net regularization Gene Regulatory Network. Please define the predictors

and the output variables, along with any additional constraints that are imposed on the model. Also, please address why this particular approach was used over other approaches developed in the literature. Providing examples of the appropriateness of this model structure for gene expression data would also be very helpful.

- On line 262, please more explicitly define what is meant by "associated with"
- On line 376, the authors state that two controls taken at the beginning and middle of the experiment were used to assess differential expression. Please further describe what is meant by "middle of the experiment". Also, this should be described in detail in the Materials and Methods section.
- A few sentences describing the metrics defined from lines 408 to 421 would be helpful. A description of how these set-based metrics are interpreted for this particular data set would be helpful.
- A description of the clustering approach used in Figure 2H in the Materials and Methods would be useful. Please also describe how it was implemented. Were only genes that were commonly differentially expressed across all experiments used? Were the resulting value (U, UD, D, or N) calculated as the average value of all genes in that set or only the gene corresponding to the Mapman bin? These details would help with the interpretation of these results.
- This reviewer is unclear of the terminology used on line 469. The line states that 50,056 ElasticNet models were generated (i.e., 6257 (note typo in manuscript) DEGs for eight stress databases). This reviewer is unfamiliar with referring to a single equation for a single DEG as a "model". The model typically refers to the collection of equations for each DEG, which was estimated using all or a subset of the data. If the terminology used in this manuscript has been used elsewhere, please provide a reference.
- On line 654, please clarify whether this is similar to what is shown in Figure 6D, just focused specifically on a single stress.
- Please further explain how the regression models were constructed for Figure 6D and Figure 7A-G. Please describe what data were used for each and highlight the relevance of the reduced R^2 values for the latter.

Minor Concerns:

- On line 464, please be explicit in referencing what the response variables (assumed to be the differentially expressed genes) and what the predictor variables are (assumed to be the differentially expressed TFs).

Additional Comments:

- In several figure captions, certain panels in the figure are not described. In Figure 2, for example, panels F) and G) are not described in the caption. Please review all figure captions to ensure that each panel is described adequately.
- On line 374, there seems to be a typo in that sentence.
- On line 225, please explicitly define what λ and α are. This reviewer assumes that these are the L1 and L2 regularization variables but this should be explicitly mentioned in the text if this is the case.
- In Figure S9, please indicate the max value of the Jaccard index in the caption.
- On line 251, please provide a reference for the Mapman ontology
- There are places in the paper that refer to 18 combinations of stresses and there are places that refer to 19 combinations of stresses. Please correct the discrepancy
- Figure 2B does not show particularly noteworthy results and may be more appropriate as a supplemental figure.
- The authors mention approaches not explicitly described in the manuscript (e.g. "UpSet plot" on line 393). A link for this was provided in the figure caption but mention of this link and/or a reference may be appropriate for the main manuscript.
- For Fig. S5, please indicate what that columns represent
- There is a typo on line 469. 6275 should be 6257.

Reviewer #2 (Remarks to the Author):

Plants are sessile and thus it is critical for them to acclimate themselves to changing environments

by appropriately adjusting gene expression profile. To understand the basic transcriptional regulatory mechanism against various abiotic stresses in plants, the authors selected the liverwort *Marchantia polymorpha* as a model, of which gene regulatory system is less redundant than those in other model plants. By comparing transcriptional profiles in *M. polymorpha* under single or combined abiotic stresses, the authors reconstructed gene regulatory networks (GRNs) specific to each stress condition and compared them, revealing that there are hierarchy in the stress GRNs and that a GRN for two stresses in combination is a simple arithmetic addition of GRNs for each of the stresses. Also revealed are transcription factors that are likely involved in regulating GRNs observed. Then the authors compared the TF-regulatory networks of *M. polymorpha* and *A. thaliana* to find minor conservation between them. Finally, the authors introduce a *Marchantia*-version of eFP browser and CoNekT database, which should help users explore the transcriptional and functional aspects of the *M. polymorpha* genome.

Some of the relevant findings of this study are the hierarchy in the stress GRNs and the additive nature of transcriptional response against abiotic stresses. Although their mechanisms and biological significance are poorly discussed, the interpretations are supported by the data provided. However, another claim they made, i.e., the stress-related TF-regulatory networks in *M. polymorpha* and *A. thaliana* are poorly conserved is less convincing. To evaluate the degree of conservation in GRN, it should be more appropriate to compare orthologous genes that are regulated by a given TF, instead of genes that have similar biological functions, because the same biological outcome can be provided by different genes or the biological outcomes of a given set of orthologs/paralogs can differentiate (neofunctionalization). Also, the claim that knowledge gained from model species such as *Arabidopsis* may not applicable to crop species is not persuasive because *A. thaliana* is by far less diverged from crop species than from *M. polymorpha*.

There are some more issues as listed below:

- Lines 317-318, 'Single stresses showed varying degrees of effect on plant growth on day 15' Supplementary Figure S1 shows considerable variations in controls (Days 15 and 21). Are the 'significant differences' observed in the experiments really significant?

- Lines 343-344, 'While plants subjected to a combination of sub-lethal heat (33°C) and high light (435 $\mu\text{Em}^{-2}\text{s}^{-1}$) died (Figure 1D)'

It is difficult to tell if HL plants were dead, judging from Figure 1D. Maybe, the correct panel to refer to is 'Figure 1C'?

- Lines 376-377, 'we used two controls taken at the beginning and the middle of the experiment' It appears that the two controls refer to Controls D2 and H2 in Table S5, but it should be explicitly described along with their conditions.

- Figure 2F and G

The representation for 'suppression' is confusing. The ratio depends on the order of two conditions to be compared, which appears oppositely in the color code. Consider showing in a triangular matrix.

- Lines 551-569

The information given in this paragraph might be useful for database contents but distracting for verifying the inferred GRNs.

- Figure 8C

Gemma cups occur only on the dorsal surface just above the midrib (Shimamura *Plant Cell Physiol.* 57:230-256, 2016). Panel 'Antheridium' should be corrected in two aspects. First, 'androcyte mother cells' should be 'spermatid mother cells', Then, the antheridium, which is a 'rugby ball'-like structure mostly consisting of spermatid mother cells, should be colored instead of the tissue and cells (jacket cells) surrounding the antheridium.

- Section starting from L681, 'eFP browser and CoNekT database for *Marchantia*'

The browser and database themselves look quite nice, but their description do not seem to be sufficiently relevant to the preceding content. It would be helpful if the authors provide steps to create figures like Figure 2H etc, for example.

There are another genome browser and co-expression databases for *M. polymorpha* in service, MarpolBase (<https://marchantia.info>) and MBEX (<https://marchantia.info/mbex/>). Describing the difference between the two systems, i.e., MarpolBase/MBEX and eFP/CoNeKT, should help readers for their choice.

- Figure 8D

I guess genes on the squares of the same color belong to the same orthogroup, as indicated below as 'Legend'. The genes in the same orthogroup are supposed to be homologous, which makes a blue dashed lines unnecessary.

Minor points

- Species names should be shown consistently (e.g., *Marchantia* or *M. polymorpha*).

- Figures should be given in better resolution, or as vector graphics.

- Figure 1B

More explanation is required for controls F, L, O, and R.

- Figure 3C

Explanation for the color-code (green, red and yellow) in the left-most column is required.

- Figure 4

Panel A, readability should be improved.

Panel B, the order of the color-code should be consistent with the graph.

Legend, Line 544, 'D)' should be 'B)'.

Conventional arrows and Ts for activation and repression, instead of color-codes, may be clearer to show the TFs' regulations (Figure S15, too).

- Figure 4 legend, L544

'D)' should be 'B)'.

- Figure S1

Labels for the vertical axis are missing in some panels. It may be simpler to remove the labels but explain in the legend. The order of the control batches seems odd: 'A, B, D, C, F,....N, L, M,...W, V, U, Q, R'.

- Figure S8A

The horizontal dashed line might be misleading. A vertical dashed line between 5 and 6 in the 'InMoreThan' axis might be more consistent with the text. Also, 'genecount' and 'tfcoutnt' can be shown as 'Gene' and 'TF', respectively, for simplicity.

- Table S7

Explanation for the meaning of the color codes is needed.

Reviewer #3 (Remarks to the Author):

The manuscript "Cross-stress gene expression atlas of *Marchantia polymorpha* reveals the hierarchy and regulatory principles of abiotic stress responses" by Qiao Wen Tan et al. comprises the transcriptomic analyses to single and combined abiotic stresses in the model liverwort *Marchantia polymorpha*. The rationale to use *Marchantia* for this work resides in its low gene redundancy, which in principle could facilitate the dissection of GRNs. One of the main conclusions of this work is the lack of conservation between the stress responses in *Arabidopsis* and *Marchantia*, which in my opinion could be further substantiated by improved analyses or description of the current approach. The authors provide convincing data on the additive effect of combined stresses, which is an interesting phenomenon given that a general stress response has been proposed in other plants such as *Arabidopsis*. Moreover, the authors incorporated their data

into two online databases (CoNekT and Marchantia eFP browser) which represent valuable tools for the Marchantia research community. The analyses performed in the manuscript are well explained and easy to follow.

I have a few concerns about the experimental design and the conclusions drawn on this work. First, I understand the difficulties of comparing the exposure and response to a range of different stresses. Have the authors considered that the plants subjected to osmotic and salt stress (which present a severe growth inhibition compared to other stresses) might show a too late or acclimated transcriptional response? I am not sure if plants grown on 100 mM mannitol or 40 mM salt represent the best conditions to perform a comparative analysis with other stresses that do not cause such growth inhibition. I don't think the experiments should be repeated with additional time points, but the authors should discuss the possibility that transcriptional changes potentially similar to other stress responses might have not been captured in the salt- and mannitol-treated plants.

Regarding the non-conservation of the stress response between Arabidopsis and Marchantia, I am not sure that the differences in GRNs based on automatic ortholog predictions suffice. Do the analyses in Fig. 2H (MapMan) indicate that the genes regulated by the different stresses are involved in similar biological pathways in different plant species? If the overall response is similar and activated by TFs from the same family (even though not by the predicted ortholog), the responses in different plants would not be considered radically different. The authors could explain more in detail the differences between the stress response in Arabidopsis and Marchantia, given that this is one of the major conclusions of their work. I do not think that the claim that the stress responses in Arabidopsis and Marchantia are different supports the idea that the knowledge obtained in Arabidopsis cannot be used to be transferred to crops (lines 819-822). Phylogenetically and morphologically, Arabidopsis and crops are more closely related than Arabidopsis and Marchantia.

The authors could explain better what are the union GRNs in Figure S9B.

It would have been nice to have some functional validation of the GRNs in Marchantia, given its enormous potential as model system (carefully explained in the introduction).

Minor points:

Even though Marchantia is considered to have a general low gene redundancy compared to Arabidopsis, this is mostly true for signaling components but not for secondary metabolism. Hence, the fact that Marchantia has several PALs is not entirely surprising. Please keep in mind that the number of Marchantia genes is roughly two thirds of the total number of Arabidopsis genes.

There is a lot of variability in the size of control plants (Fig. S1). Moreover, the 50 % growth inhibition of plants grown for 3 days in dark conditions is difficult to appreciate in Fig. 1B. Is this phenotype reproducible?

Line 405: Please, check the references to the figures

Line 500: Not in bottom row of S10, please correct

TFs families in Figure 3 and S10 should have the same colors for clarity.

Dear Reviewers,

Our responses are in blue

The text that is found in the manuscript is *italic*, while any changes are in *red*, e.g.: ‘*To support this observation, we found that heat...*’

The quality of the figures in this document is lower than the original figures, due to conversion to PDF. To see the high resolution figures, please take a look at the uploaded pdfs.

Best wishes,
Marek

REVIEWER COMMENTS

Reviewer #1 (Remarks to the Author):

SUMMARY:

The authors analyze data obtained from exposing the plant, *Marchantia polymorpha*, to seven individual stresses (heat, cold, salt, osmotic, light, dark, and nitrogen deficiency) and to 19 pairwise combinations of those stresses. The constructed an abiotic gene expression atlas of *Marchantia* to understand how plant integrate and respond to the various environmental cues. Genes that were found to be differentially expressed against controls from two different batches were considered for further analysis. DEGs expressed in 5 or more experiments were used to calculate gene regulatory networks (GRN) for all experiments and stress-specific networks using a subset of experiments that included the respective experiments. GRN were generated using linear regression with elastic net regularization. The authors identified 75 robustly-responding TFs, revealing that dominant stresses (e.g. heat and darkness) differentially express a higher number of TFs than the non-dominant stresses. The inferred GRNs showed that the TFs active in the dominant stresses regulate more genes and other TFs than TFs from non-dominant stresses. These regression GRNs also showed that gene expression changes observed from pairwise experiments could be explained by a linear combination of the log₂ fold change values from the single experiments. To make the resulting data more accessible, they provide an eFP browser for *Marchantia*.

REVIEW:

What are the noteworthy results?

The primary noteworthy result from the manuscript is the rich dataset capturing the transcriptional response for single and combinatorial stresses. The process for identifying relevant specifications for the single and combinatorial stresses experiments were systematic and produced a rich dataset that would be of interest to other researchers in the field.

The result identifying that combinatorial stresses can be described as linear combinations of the results from single stresses is particularly noteworthy. This tracks with the low level of redundancy seen in *Marchantia*. Also, the identification of a hierarchy of stress responses is also noteworthy.

Major Concerns:

- N/A

Minor Concerns:

- N/A

Will the work be of significance to the field and related fields? How does it compare to the established literature? If the work is not original, please provide relevant references.

The resulting data generated, along with the web interface, would be of significance to researchers in the field. This is a rather expansive data set that could be used for further analysis. The results here could also be further compared to combinatorial stress response results from other plants such as maize and Arabidopsis to better understand how these plants have evolved to handle stresses typically seen in nature.

Response: We thank the reviewer for the thorough and constructive comments.

Major Concerns:

- N/A

Minor Concerns:

- N/A

Does the work support the conclusions and claims, or is additional evidence needed?

It seems like the work broadly supports the major conclusions and claims. Further description and refinement of the analysis and methodology would help to strengthen the manuscript.

Major Concerns:

- N/A

Minor Concerns:

- N/A

Are there any flaws in the data analysis, interpretation and conclusions? Do these prohibit publication or require revision?

There are significant flaws in the interpretation of some of the results. Please see below for specific points.

Major Concerns:

1.1 On lines 422 to 432, a more formal interpretation beyond the heatmap shown in Figure 2 F and G

would be appropriate. For example, is it fair to say that the similarity metric ranged from 5% similarity to 25% similarity. A quantitative assessment and interpretation would be useful.

Response: We agree that these figure panels and the corresponding text should be improved. To address this issue, we have added numbers to these heatmaps, and also discuss the employed metrics:

Stresses showing the highest similarity in terms of DEG responses comprise salt and mannitol for up- and down-regulated genes (Figure 2C, *similarity values for SM is 0.27 and 0.20, respectively*), salt and nitrogen deficiency for upregulated genes (Figure 2C, *similarity value 0.21*), and heat and darkness for downregulated genes (Figure 2D, *similarity value 0.26*). The *set analysis for novel interactions* revealed that the salt+mannitol combination *involved* DEGs that were not found in the individual stresses (Figure 2E-F, *novel interaction value 0.48 and 0.56, for up- and downregulated genes*), suggesting that the two stresses can activate altogether different responses when combined. *Lastly, the suppression analysis showed that darkness is a strong suppressor for many stresses (negative suppression scores ranging from -0.21 to -0.65 when D is Sx, Figure 2E-F), except HD (-0.09 and -0.06 for up- downregulated genes, Figure 2E-F), indicating that heat stress and darkness are comparably dominant. To support this observation, we found that heat stress could suppress other stresses (negative suppression scores ranging from -0.01 to -0.49 when H is Sx. Figure 2E-F).*

1.2 Based on the explanation in the manuscript, it seems like the results in Figure S9A is quantify the genes and their corresponding equation that has an R^2 value above a certain amount. Thus, setting a filter of $R^2 > 0.8$ indicates the genes where there is some level of confidence in the prediction. This is a different interpretation than what is shown in the Figure, which refers to Number of models. The use of the term “Number of models” infers that the approach was ran N number of times and from these N runs, these are the resulting models (with each model being comprised of the equations for all DEGs) with R^2 values $>$ some value. The interpretation of these results should be revised.

Response: We agree that the word ‘models’ might be misunderstood. To clarify, we provided the following explanation in the opening of the ‘**Identification of high-confidence transcription factors involved in stress response**’:

‘To infer the stress-responsive GRN (Figure S8B), we used ElasticNet, which allows building a regression model that can predict gene expression of a gene (response variable), given the expression of transcription factors (predictor variables).’

Furthermore, we elaborate on the interpretation of Figure S9A. To make the figure and text more clear, we renamed the y-axis “number of DEGs with models of at least R^2 ”:

‘To study the performance of the models across the different expression datasets, we investigated the number of DEGs (response variables) for which the corresponding models showed coefficient of determination (R^2) greater than or equal to a given value. Interestingly, the number of DEGs for which reliable models ($R^2 > 0.8$) could be obtained was larger when using the compilation of experiments that shared a stress, rather than when using all data (Figure S9A).’

1.3 There are places in the manuscript where the explanation of the results are confusing and or does not flow in a way that is easily followed. This is a result of the use of terminology not common in the field (see above comment regarding the reference of each DEG equation as a model) or the extrapolation of results from several figures without a clear connection. This is a major flaw and should be clarified.

Response: We thank the reviewer for raising this concern. To address this, we have asked our colleagues to read the manuscript carefully and highlight any unclear parts. As seen in several places in the manuscript, we have made changes to clarify the text.

Minor Concerns:

- N/A

Is the methodology sound? Does the work meet the expected standards in your field?

See below.

Major Concerns:

1.4 The overall methodology for modeling of the GRN could be improved. While using these results to get a broad understanding of the linear relationship between TFs and potential target genes are noteworthy, this approach does not capture the state-of-art in building gene regulatory networks for biological systems. While the development of new approaches for building GRNs is not the goal of the manuscript, further justification for the methodology used is warranted. The manuscript could also benefit from a description of the pros and cons associated with using such an approach.

Response: The gene expression values were log-transformed and scaled prior to modeling; hence, the relationship between a DEG, as a response gene, and a TFs is non-linear and represented, in absolute counts, as a product. This transformation is common and is applied in prominent approaches for reconstruction of GRNs based on regularized regressions. For instance, the inferelator approach (as first in the series) along with LASSO and extensions thereof has been shown to outperform other approaches on the DREAM5 challenge. We are aware that random forests have recently been used

with success to reconstruct GRNs, but they do not provide an easy way for model comparison – which is one of the reasons why we opted to use regularized regressions without compromising model performance.

We now mention this in M&M:

‘The gene expression values were log-transformed and scaled prior to modeling; hence, the relationship between the expression of a DEG, as a response variable, and the expression of TFs, as predictor variables, is non-linear.’

And the main text:

‘We chose ElasticNet, since regularized regressions achieved competitive performance when compared to other GRN inference methods⁴³, and unlike some of the top performing methods (e.g., Genie3⁴⁴), facilitate easier comparison of inferred interactions across different datasets.’

Minor Concerns:

- N/A

Is there enough detail provided in the methods for the work to be reproduced?

Significant improvement in the description of the computational methods is needed. See below comments for detailed recommendations.

Major Concerns:

1.5 Please further describe in the Materials and Methods section the construction of the linear regression with elastic net regularization Gene Regulatory Network. Please define the predictors and the output variables, along with any additional constraints that are imposed on the model. Also, please address why this particular approach was used over other approaches developed in the literature. Providing examples of the appropriateness of this model structure for gene expression data would also be very helpful.

Response: We thank the reviewer for pointing out the lapse in the description of methods. We have updated the section to better describe the reconstruction of the gene regulatory network.

‘The GRN was reconstructed using linear regression as described in Eq. (1) where the response variable, Y , represents the vector of (log-transformed) expression levels of a DEG; the predictor variables, X , represents a matrix of (log-transformed) expression levels of the TFs; and the error term, ε , of normally distributed variance. The parameters (i.e regression coefficients), β , were estimated by minimizing the sum of the loss and a weighted combination of the first and second norm of the regression coefficients, termed ElasticNet regularization³⁴ as described in Eq. (2), which provided a good compromise between model sparsity (i.e., feature selection corresponding to the inclusion of transcription factors in the model) and model explanatory power.

$$Y = X\beta + \varepsilon \tag{1}$$

$$L_{enet}(\hat{\beta}) = \frac{\sum_{i=1}^n (y_i - x_i' \hat{\beta})^2}{2n} + \lambda \left(\frac{1-\alpha}{2} \sum_{j=1}^m \hat{\beta}_j^2 + \alpha \sum_{j=1}^m |\hat{\beta}_j| \right) \tag{2}$$

Three- and five-fold cross-validation were used for the stress-specific data and all data, respectively, to determine the optimal λ (L1 regularization; 0.1 to 0.9, 0.1 steps) and α (L2 regularization; 0, 0.001, 0.01, 0.05, 0.1, 0.5, 1, 1.5, 2, 10, 100) that resulted in the lowest coefficient of variation for each model. We filtered for good quality models with $R^2 > 0.8$ (Figure S9A).'

We now mention the reason for using ElasticNet over other approaches:

'We chose ElasticNet, since regularized regressions achieved competitive performance when compared to other GRN inference methods⁴³, and unlike some of the top performing methods (e.g., Genie3⁴⁴), facilitate easier comparison of inferred interactions across different datasets.'

1.6 On line 262, please more explicitly define what is meant by "associated with"

Response: We have clarified the association to explicitly mention the regulatory relationships found between Mapman bin and TF in the following text:

'Robustly-responding MapMan bins (Figure S13) were considered to be regulated by robustly responding TFs if at least 5% of the genes (Figure S14) in the MapMan bin were regulated by the TF in our GRN (Figure 4C, D and S15).'

1.7 On line 376, the authors state that two controls taken at the beginning and middle of the experiment were used to assess differential expression. Please further describe what is meant by "middle of the experiment". Also, this should be described in detail in the Materials and Methods section.

Response: We have updated the Materials and Methods section and the text to explain the selection of the two controls:

'Due to equipment constraints, the experiment was carried out over 22 batches. Some variability in size was observed in the controls (Figure S1). To account for this variability, we have sequenced controls from batches F and L to impose a more stringent criteria for the identification of differential gene expression.'

In the results, we addressed this with:

'we first identified differentially expressed genes in the single stresses and the 18 combinations of two stresses. For more robust inferences, we used two controls (grown on half-strength Gamborg B-5 Basal agar at 24°C under continuous LED light at 60 $\mu\text{Em}^{-2}\text{s}^{-1}$) taken from batches F and L and consider'

1.8 A few sentences describing the metrics defined from lines 408 to 421 would be helpful. A description of how these set-based metrics are interpreted for this particular data set would be helpful.

Response: We have furnished an example of the interpretation of cold-dark stress in the main text to support the interpretation of the set metrics:

'Using the comparison of upregulated genes in cold+dark as an example (Figure 2C), the similarity between upregulated genes in cold and dark is low (white field, 0.04). Furthermore, cold+dark combination showed a comparably high proportion of novel interactions (medium blue, 0.37),

indicating that the combined cold+dark stress upregulates genes not found in either cold or dark stress. Lastly, the suppression analysis (Figure 2E) revealed a positive value (light red field, 0.22), indicating that there is a high proportion of genes that are upregulated during cold, but not upregulated in cold+dark. This indicates that darkness suppresses part of the upregulated cold stress response.'

1.9 A description of the clustering approach used in Figure 2H in the Materials and Methods would be useful. Please also describe how it was implemented. Were only genes that were commonly differentially expressed across all experiments used? Were the resulting value (U, UD, D, or N) calculated as the average value of all genes in that set or only the gene corresponding to the Mapman bin? These details would help with the interpretation of these results.

Response: We thank the reviewer for highlighting the lack of description here and have furnished the following in the Materials and Methods:

'Identification of transcription factors and differentially expressed pathways

'To identify which stresses (columns) and biological pathways (rows) were similar, we first calculated the Jaccard distance (JD) between all columns/rows. The JD values were used to build a condensed distance matrix which was used as input for the hierarchical clustering algorithm.'

1.10 This reviewer is unclear of the terminology used on line 469. The line states that 50,056 ElasticNet models were generated (i.e., 6257 (note typo in manuscript) DEGs for eight stress databases). This reviewer is unfamiliar with referring to a single equation for a single DEG as a "model". The model typically refers to the collection of equations for each DEG, which was estimated using all or a subset of the data. If the terminology used in this manuscript has been used elsewhere, please provide a reference.

Response: We hope that the clarification provided above makes it more clear (we have corrected the typo):

'To infer the stress-responsive GRN (Figure S8B), we used ElasticNet, which allows building a regression model that can predict gene expression of a gene (response variable), given the expression of transcription factors (predictor variables).'

1.11 On line 654, please clarify whether this is similar to what is shown in Figure 6D, just focused specifically on a single stress.

Response: This section pertains to the regression analysis of single genes, rather than averages (Figure 6D). We have further clarified the differences between the regression results shown in Figure 6D and Figure 7A-G:

'To further examine how well S_{xy} can be explained by the \log_2fc values of the individual genes, rather than averages (Figure 6C) from the perspective of various stresses, we performed another 3-dimensional linear regression (Figure 7A-G), where the x-axis (S_x) and y-axis (S_y) of the cold-centric (Figure 7A) plot represents individual gene expression values from cold stress and all other stresses respectively. While the R^2 values dropped, which is not unexpected as the first analysis in Figure 6D was done on averaged \log_2fc values which likely smoothed out gene-specific variations in gene expression, the resulting models could still approximate the expected values well ($R^2 = 0.57-0.67$).'

1.12 Please further explain how the regression models were constructed for Figure 6D and Figure 7A-G. Please describe what data were used for each and highlight the relevance of the reduced R^2 values for the latter.

Response: We updated the material and methods to describe how the regression models were constructed for Figures 6D and 7A-G.

‘Linear regression of gene expression

Trends in gene expression during combined stress were revealed using ordinary least squares (OLS) regression. For the general trend, we first grouped the expression values (\log_2fc) according to the 9 possible combinations of responses in S_x and S_y (i.e., downregulated/upregulated/no change in S_x and S_y) for each stress, where S_x represents the stress of interest. The \log_2fc values were then averaged. In addition, linear regression shown in Figure 7A-G was performed on non-averaged gene expression values for each stress group independently. For example, for cold stress, this will involve S_{xy} of CD, CL, CN, CS and CM, where S_x is C and S_y are all other corresponding stresses. A standard linear regression of the (averaged) \log_2fc was performed using the scikit-learn linear model package (`linear_model`) with S_{xy} as the response variable and S_x and S_y as the predictor variables.’

We have included a description on the significance of the decreased R^2 values in the subsequent analyses.

‘While the R^2 values dropped, which is not unexpected as the first analysis in Figure 6D was done on averaged \log_2fc values which likely smoothed out gene-specific variations in gene expression, the resulting models could still approximate the expected values well ($R^2 = 0.57-0.67$).’

Minor Concerns:

1.13 On line 464, please be explicit in referencing what the response variables (assumed to be the differentially expressed genes) and what the predictor variables are (assumed to be the differentially expressed TFs).

Response: We modified the writing to be more explicit:

‘To infer the stress-responsive GRN (Figure S8B), we used ElasticNet, which allows building a regression model that can predict gene expression of a gene (response variable), given the expression of transcription factors (predictor variables).’

Additional Comments:

1.14 In several figure captions, certain panels in the figure are not described. In Figure 2, for example, panels F) and G) are not described in the caption. Please review all figure captions to ensure that each panel is described adequately.

Response: We thank the reviewer for spotting this. We have amended the figures and the figure legends. For example, several panels in Figure 2 are changed, and we have amended the captions accordingly.

1.15 On line 374, there seems to be a typo in that sentence.

Response: We have edited the sentence:

*'To better understand how **Marchantia** responds to stresses, we first identified differentially expressed genes in the single stresses and the 18 combinations of two stresses.*

1.16 On line 225, please explicitly define what λ and α are. This reviewer assumes that these are the L1 and L2 regularization variables but this should be explicitly mentioned in the text if this is the case.

Response: We have clarified this with the following edits:

'Three- and five-fold cross-validation were used for the stress-specific data and all data, respectively, to determine the optimal λ (L1 regularization; 0.1 to 0.9, 0.1 steps) and α (L2 regularization; 0, 0.001, 0.01, 0.05, 0.1, 0.5, 1, 1.5, 2, 10, 100) that resulted in the lowest coefficient of variation for each model. We filtered for good quality models with $R^2 > 0.8$ (Figure S9A).'

1.17 In Figure S9, please indicate the max value of the Jaccard index in the caption.

Response: The maximum observed value for Jaccard index is indicated in the updated figure caption: *'Maximum value of Jaccard index was observed in 'Union (1)' at 0.019.'*

1.18 On line 251, please provide a reference for the Mapman ontology

Response: We have added the reference to Mapman ontology:

'To understand how specifically regulated transcription factors might be affecting certain biological processes, we took transcription factors and second-level Mapman bins³² that'

1.19 There are places in the paper that refer to 18 combinations of stresses and there are places that refer to 19 combinations of stresses. Please correct the discrepancy

Response: We have edited the following sentence in line 349 to clarify the discrepancy:

'The resulting panel of 7 single stresses, 18 combinations of two stresses (excluding heat-light that resulted in plant death)'

1.20 Figure 2B does not show particularly noteworthy results and may be more appropriate as a supplemental figure.

Response: We agree to move it as supplementary Figure 4B.

1.21 The authors mention approaches not explicitly described in the manuscript (e.g. "UpSet plot" on line 393). A link for this was provided in the figure caption but mention of this link and/or a reference may be appropriate for the main manuscript.

Response: We now include the references for UpSet plot in the main text:

'We used the UpSet plot⁴² to elucidate these similarities, which shows the intersection of multiple sets for upregulated (Figure S5A) and downregulated (Figure S5B) genes.'

1.22 For Fig. S5, please indicate what that columns represent

Response: Updated the figure legend for clarity:

'Rows represent the various stress conditions and columns represent the genes that are found in the particular group of sets indicated by the dots and connecting lines. The UpSet plot package is available from <https://upsetplot.readthedocs.io/en/stable/>.'

1.23 There is a typo on line 469. 6275 should be 6257.

Response: We thank the reviewer for spotting the typo:

'~~62576275~~ DEGs for eight stress datasets'

We would like to thank the reviewer for these constructive comments.

Reviewer #2 (Remarks to the Author):

Plants are sessile and thus it is critical for them to acclimate themselves to changing environments by appropriately adjusting gene expression profile. To understand the basic transcriptional regulatory mechanism against various abiotic stresses in plants, the authors selected the liverwort *Marchantia polymorpha* as a model, of which gene regulatory system is less redundant than those in other model plants. By comparing transcriptional profiles in *M. polymorpha* under single or combined abiotic stresses, the authors reconstructed gene regulatory networks (GRNs) specific to each stress condition and compared them, revealing that there are hierarchy in the stress GRNs and that a GRN for two stresses in combination is a simple arithmetic addition of GRNs for each of the stresses. Also revealed are transcription factors that are likely involved in regulating GRNs observed. Then the authors compared the TF-regulatory networks of *M. polymorpha* and *A. thaliana* to find minor conservation between them. Finally, the authors introduce a *Marchantia*-version of eFP browser and CoNekT database, which should help users explore the transcriptional and functional aspects of the *M. polymorpha* genome.

Some of the relevant findings of this study are the hierarchy in the stress GRNs and the additive nature of transcriptional response against abiotic stresses. Although their mechanisms and biological significance are poorly discussed, the interpretations are supported by the data provided.

2.1 However, another claim they made, i.e., the stress-related TF-regulatory networks in *M. polymorpha* and *A. thaliana* are poorly conserved is less convincing. To evaluate the degree of conservation in GRN, it should be more appropriate to compare orthologous genes that are regulated by a given TF, instead of genes that have similar biological functions, because the same biological outcome can be provided by different genes or the biological outcomes of a given set of orthologs/paralogs can differentiate (neofunctionalization).

Response: We agree with the reviewer that these analyses should be strengthened. Consequently, we have expanded the section on the comparison of *Marchantia* and *Arabidopsis* TF gene families. In addition, we have included the discussion regarding Figure S9 to account for the comparison of genes regulated by the orthologs of *Arabidopsis* and *Marchantia*. Finally, based on the comments of the reviewer and reviewer 3, we have expanded the analysis by looking at the differentially expressed gene families in *Marchantia* and *Arabidopsis*. The analysis revealed a statistically significant conserved response between the two species (Please see below). Consequently, we removed the strong statement claiming no conservation between the two species.

*'Typically, each **Marchantia** TF has many **Arabidopsis** orthologs due to larger gene families in **Arabidopsis**, and the functions of the **Arabidopsis** orthologs are annotated based on evidence from literature or gene expression from the eFP browser abiotic stress dataset³⁷ (Table S8). To summarize the findings, we grouped the **Marchantia** TFs according to the stress-specific occurrence and counted the number of literature and gene expression evidence in the corresponding **Arabidopsis** orthologs (Figure 5A, Table S9).'*

Finishing sentences:

'The analysis revealed that for all stresses, with the exception of dark, the number of similar gene expression families that are differentially expressed in the stresses is higher than expected by chance (Figure 5D, the observed JI is a black dot, significantly higher JI values are indicated by black

asterisk). This indicates that despite the seemingly different responses (Figure 5A-C), the two plants differentially express a similar set of gene families (Figure 5D).'

We also made further changes to the abstract and discussion (please see our replies and analysis to reviewer 3).

2.2 Also, the claim that knowledge gained from model species such as Arabidopsis may not applicable to crop species is not persuasive because *A. thaliana* is by far less diverged from crop species than from *M. polymorpha*.

Response: We reviewed the claim and agree with the reviewer that the claim is unfair and revised the statement:

'This suggests that each model plant uses a different strategy to acclimate to stress, and considerations regarding the phylogenetic distance between model species and crops of interest should be accounted for when attempting to make knowledge transfers from a model plant to crops.'

There are some more issues as listed below:

2.3 Lines 317-318, 'Single stresses showed varying degrees of effect on plant growth on day 15' Supplementary Figure S1 shows considerable variations in controls (Days 15 and 21). Are the 'significant differences' observed in the experiments really significant?

Response: We agree with the reviewer that there are some variations between the sizes controls. However, the comparisons were also made with reference to the batch controls (Figure S1), and we observed that the gene expression differences between controls were minor.

'Single stresses showed varying degrees of effect on plant growth on day 15 (day of harvest when compared to their respective batch controls, Figure S1 shows growth measurements against respective batch controls, Supplemental Data 1 shows agar plates, Figure 1A shows growth measurements against controls averaged across all batches) and day 21 (growth measurements for 6 days post-stress for environmental stresses against controls averaged across all batches, Figure 1B).'

2.4 Lines 343-344, 'While plants subjected to a combination of sub-lethal heat (33°C) and high light (435 $\mu\text{Em}^{-2}\text{s}^{-1}$) died (Figure 1D)' It is difficult to tell if HL plants were dead, judging from Figure 1D. Maybe, the correct panel to refer to is 'Figure 1C'?

Response: We apologize and thank the reviewer for spotting the mistake. It has since been corrected: *'high-light (435 $\mu\text{Em}^{-2}\text{s}^{-1}$) died (Figure 1C)'*

2.5 Lines 376-377, 'we used two controls taken at the beginning and the middle of the experiment' It appears that the two controls refer to Controls D2 and H2 in Table S5, but it should be explicitly described along with their conditions.

Response: We thank the reviewer for the suggestion and have updated the main text and added description in Table S5 for clarity:

'For more robust inferences, we used two controls (grown on half-strength Gamborg B-5 Basal agar at 24°C under continuous LED light at 60 $\mu\text{Em}^{-2}\text{s}^{-1}$) taken from batches F and L'

2.6 Figure 2F and G

The representation for 'suppression' is confusing. The ratio depends on the order of two conditions to be compared, which appears oppositely in the color code. Consider showing in a triangular matrix.

Response: This is a great suggestion, please find the triangular matrices capturing suppression for up- and down-regulated genes. We have modified the text to accommodate the panels:

In legend of Figure 2:

'Suppression analysis for E) upregulated and F) downregulated genes. A darker shade of red and blue indicates that more genes from the first (Sx) and second (Sy) stress are not represented in the combined stress (Sxy), respectively.'

2.7 Lines 551-569

The information given in this paragraph might be useful for database contents but distracting for verifying the inferred GRNs.

Response: We regret the confusion caused by this section. The verification of inferred GRN has been covered in the previous paragraphs. To resolve this, we decided to add the following header:

'Investigating the regulatory wiring of stress-responsive transcription factors and biological processes'

2.8 Figure 8C

Gemma cups occur only on the dorsal surface just above the midrib (Shimamura Plant Cell Physiol. 57:230-256, 2016). Panel 'Antheridium' should be corrected in two aspects. First, 'androcyte mother cells' should be 'spermatid mother cells', Then, the antheridium, which is a 'rugby ball'-like structure mostly consisting of spermatid mother cells, should be colored instead of the tissue and cells (jacket cells) surrounding the antheridium.

Response: We thank the reviewer for spotting this. We have updated the database and the figure panel.

2.9 Section starting from L681, 'eFP browser and CoNekT database for Marchantia'

The browser and database themselves look quite nice, but their description do not seem to be sufficiently relevant to the preceding content. It would be helpful if the authors provide steps to create figures like Figure 2H etc, for example.

Response: We appreciate the reviewer's suggestion. The database was designed to be more open ended for the purpose of data exploration. Indeed, having a tool to do such an analysis would be highly valuable and we will take this into account for our next database update. Please see the updated Materials and Methods to see how to make Figure 2G.

Material and Methods: *'To identify which stresses (columns) and biological pathways (rows) were similar, we first calculated the Jaccard distance (JD) between all columns/rows. The JD values were used to build a condensed distance matrix which was used as input for the hierarchical clustering algorithm.'*

2.10 There are another genome browser and co-expression databases for *M. polymorpha* in service, MarpolBase ([<https://marchantia.info><<https://smex-ctp.trendmicro.com:443/wis/clicktime/v1/query?url=https%3a%2f%2fmarchantia.info&umid=91cbc1a9-ad96-435f-956a-59f8ffb64ca3&auth=682db514602deaff239ec316aa24e93dba6f746c-847713cdc969b8fd7ea131875400361ff94ecbc1>]<<https://marchantia.info><<https://smex-ctp.trendmicro.com:443/wis/clicktime/v1/query?url=https%3a%2f%2fmarchantia.info&umid=91cbc1a9-ad96-435f-956a-59f8ffb64ca3&auth=682db514602deaff239ec316aa24e93dba6f746c-847713cdc969b8fd7ea131875400361ff94ecbc1>>)] and MBEX (<https://marchantia.info/mbex/><<https://smex-ctp.trendmicro.com:443/wis/clicktime/v1/query?url=https%3a%2f%2fmarchantia.info%2fmbex%2f&umid=91cbc1a9-ad96-435f-956a-59f8ffb64ca3&auth=682db514602deaff239ec316aa24e93dba6f746c-847713cdc969b8fd7ea131875400361ff94ecbc1>>)]

0a82e624781ad1ee59f8a407bdd8ee17015dd2d5>). Describing the difference between the two systems, i.e., MarpolBase/MBEX and eFP/CoNekT, should help readers for their choice.

Response: We thank the reviewer for bringing this up and have included the two Marchantia databases in the discussion.

‘Notably, there are two other databases dedicated to Marchantia, MarpolBase³⁰ and MarpolBaseExpression (MBEX)⁴⁸, which are genome and expression databases respectively. With various information and functionality spread across databases, we summarize the main functionalities for the databases and the usefulness of each for different purposes. Marpolbase is the place to go for the download of genome-related information and tools such as BLAST to Marchantia and other plant species of interest, guide RNA design for CRISPR/Cas9, and other tools that aid in the retrieval of gene-related information. On the other hand, MBEX, eFP, and CoNekT are expression-based databases. Among the three, eFP serves purely as a visualization of expression data in the anatomy of the plant, and the Marchantia instance is one of the many species found in the database collection. Visualization by anatomy is also available on MBEX as ‘Chromatic Expression Images’. While MBEX and CoNekT overlap in the area of expression and co-expression tools, MBEX provides more downstream analysis tools that allow for the analysis of differential expression, enrichment of biological functions, and set relations between conditions in Marchantia. On the other hand, CoNekT provides an organ-centric dissection of the expression data with the integration of orthology information across 12 other species from the Viridiplantae and tools for comparison across species and tissues.’

2.11 Figure 8D

I guess genes on the squares of the same color belong to the same orthogroup, as indicated below as 'Legend'. The genes in the same orthogroup are supposed to be homologous, which makes a blue dashed lines unnecessary.

Response: We thank the reviewer for pointing this out and agree that it is unnecessary here. However, this is an established feature of the CoNekT database (published in 2018) and while it may seem redundant in this publication, it does help to a certain extent in the database as the nodes are not lined up neatly in the default graph layout. We would like to keep the dashed lines, to exemplify a typical output from the database.

Minor points

2.12 Species names should be shown consistently (e.g., Marchantia or M. polymorpha).

Response: We have fixed the irregularities in species names for Marchantia and Arabidopsis in text.

2.13 Figures should be given in better resolution, or as vector graphics.

Response: We apologize for the low resolution in the compiled PDF, which happened when the manuscript was processed by the Nature Communications platform. The original figures are in high resolution PDFs.

2.14 Figure 1B

More explanation is required for controls F, L, O, and R.

Response: We expanded the explanation for the controls and linked it to the supplementary materials for more information.

Figure 1 legend: '*Representatives of control from batches F and L for independent stresses performed over 14 batches in panel B (Figure S1, Supplementary Data 1). C) Phenotypes of plants on day 21 for combined stresses. Representatives of control from batches O and R for combined stresses performed over 9 batches in panel C (Figure S1, Supplementary Data 1).*'

2.15 Figure 3C

Explanation for the color-code (green, red and yellow) in the left-most column is required.

Response: We have furnished the legend with the missing information.

Figure 3 legend: '*Green, red and yellow colors on the leftmost column of the plot represent the most commonly observed relationship between TF and gene, which corresponds to activator, repressor and ambiguous (positive and negative coefficients observed for the same TF-gene pair in different stress-specific networks) respectively. TFs are colored according to their TF family for TFs that are represented at least thrice.*'

2.16 - Figure 4

Panel A, readability should be improved.

Panel B, the order of the color-code should be consistent with the graph.

Legend, Line 544, 'D)' should be 'B)'.

Conventional arrows and Ts for activation and repression, instead of color-codes, may be clearer to show the TFs' regulations (Figure S15, too).

Response: We thank the reviewer for these suggestions. We have made the following changes:

Panel A: We believe this would be resolved with the original PDF for figures that were submitted.

Panel B: The labelling of the legend has been fixed. Color in figure legend re-ordered. Arrows and Ts for GRN: Fixed in 4A, C, D and S15.

2.17 Figure 4 legend, L544 'D)' should be 'B)'.
 Response: Corrected.

2.18 Figure S1

Labels for the vertical axis are missing in some panels. It may be simpler to remove the labels but explain in the legend. The order of the control batches seems odd: 'A, B, D, C, F,...N, L, M,...W, V, U, Q, R'.

Response: We have amended the figure, e.g.:

2.19 Figure S8A

The horizontal dashed line might be misleading. A vertical dashed line between 5 and 6 in the 'InMoreThan' axis might be more consistent with the text. Also, 'genecount' and 'tfcount' can be shown as 'Gene' and 'TF', respectively, for simplicity.

Response: We appreciate the suggestions and have updated the figure accordingly.

2.20 Table S7

Explanation for the meaning of the color codes is needed.

Response: We have updated the table to include the meaning of the color codes.

We would like to thank the reviewer for the great comments!

Reviewer #3 (Remarks to the Author):

The manuscript “Cross-stress gene expression atlas of *Marchantia polymorpha* reveals the hierarchy and regulatory principles of abiotic stress responses” by Qiao Wen Tan et al. comprises the transcriptomic analyses to single and combined abiotic stresses in the model liverwort *Marchantia polymorpha*. The rationale to use *Marchantia* for this work resides in its low gene redundancy, which in principle could facilitate the dissection of GRNs. One of the main conclusions of this work is the lack of conservation between the stress responses in *Arabidopsis* and *Marchantia*, which in my opinion could be further substantiated by improved analyses or description of the current approach. The authors provide convincing data on the additive effect of combined stresses, which is an interesting phenomenon given that a general stress response has been proposed in other plants such as *Arabidopsis*. Moreover, the authors incorporated their data into two online databases (CoNekT and *Marchantia* eFP

browser) which represent valuable tools for the *Marchantia* research community. The analyses performed in the manuscript are well explained and easy to follow.

Response: We thank the reviewer for the constructive comments and helpful suggestions.

3.1 I have a few concerns about the experimental design and the conclusions drawn on this work. First, I understand the difficulties of comparing the exposure and response to a range of different stresses. Have the authors considered that the plants subjected to osmotic and salt stress (which present a severe growth inhibition compared to other stresses) might show a too late or acclimated transcriptional response? I am not sure if plants grown on 100 mM mannitol or 40 mM salt represent the best conditions to perform a comparative analysis with other stresses that do not cause such growth inhibition. I don't think the experiments should be repeated with additional time points, but the authors should discuss the possibility that transcriptional changes potentially similar to other stress responses might have not been captured in the salt- and mannitol-treated plants.

Response: We agree with the reviewer and have highlighted the rationale behind our experimental design in the discussion:

'Notably, the chronic stresses dark, salt, osmotic, and nitrogen deficiency cause growth inhibition largely due to the longer duration of these stresses, likely capturing late-stage responses. During the comparison of chronic stresses with the acute stresses of heat, cold, and light, we may miss the effects of early-stage responses that might have been present in the acute stresses. However, we assume that the sampling of 1 day after stress induction for acute stresses captures mostly late-stage responses, allowing us to compare the chronic and acute stresses⁵³.'

3.2 Regarding the non-conservation of the stress response between *Arabidopsis* and *Marchantia*, I am not sure that the differences in GRNs based on automatic ortholog predictions suffice. Do the analyses in Fig. 2H (MapMan) indicate that the genes regulated by the different stresses are involved in similar biological pathways in different plant species? If the overall response is similar and activated by TFs from the same family (even though not by the predicted ortholog), the responses in different plants would not be considered radically different. The authors could explain more in detail the differences between the stress response in *Arabidopsis* and *Marchantia*, given that this is one of the major conclusions of their work.

Response: We thank the reviewer for bringing this up. We have downloaded differential gene expression data for *Arabidopsis*, and as the reviewer suggested, we took a look at gene families, rather than orthologs. The analysis caused us to revise our conclusion:

Abstract: *'While the transcriptomic responses showed a conserved differential gene expression between Arabidopsis and Marchantia, we also observed a strong functional and transcriptional divergence between the two species.'*

Introduction: *'Comparing these TFs to gene expression responses and biological function of Arabidopsis thaliana orthologs revealed poor agreement between the two plants. However, when we looked at the profile of differentially expressed genes in Marchantia and Arabidopsis homologous gene families, there were significant similarities in cold, heat and salt stress based on the Jaccard index, suggesting conservation of TF responses in Marchantia and Arabidopsis at the gene family level.'*

Results: *'To investigate whether the differential gene expression patterns of stress responses are similar between Marchantia and Arabidopsis, we downloaded the differentially expressed genes for Arabidopsis cold, dark, heat and salt¹⁶. Next, we calculated the Jaccard Index (JI) similarity of the gene families that are differentially expressed between Arabidopsis and Marchantia genes. The analysis revealed that for all stresses, with the exception of dark, the number of similar gene expression families that are differentially expressed in the stresses is higher than expected by chance (Figure 5D, the observed JI is a black dot, significantly higher JI values are indicated by black asterisk). This indicates that despite the seemingly different responses (Figure 5A-C), the two plants differentially express a similar set of gene families (Figure 5D).'*

Figure 5 legend: *'D) Similarity in differential gene expression between Marchantia and Arabidopsis homologs belonging to the same gene family. The black points indicate the observed JI capturing the similarity of the differentially expressed gene families. The violin plots indicate the JI where the gene-family assignments have been shuffled 1000 times. The black asterisk shows cases where the observed JI is significantly higher (p-value < 0.05).'*

Material and methods: *'To investigate the behavior of homologous TFs in Arabidopsis and Marchantia during various stress conditions, we compared the occurrence of DEGs from our dataset and from Ferrari et. al. ¹⁶ study on Arabidopsis cold (E-GEOD-63406), dark (E-GEOD-67956), heat (E-GEOD-72806), and salt stresses (E-GEOD-72806). Corrected p-value (< 0.05) and an absolute log₂fc cut off of more than 1 were used to determine DEGs for both datasets. Homologous relationships between*

Arabidopsis and *Marchantia* were obtained from PLAZA Dicots 5.0³⁶ and the compatibility of gene names from *Marchantia* genome v3 to v5.1r1 was ensured using the conversion table on MarpolBase³⁰. For the analysis, only homolog IDs common to both species were used. The Jaccard Index (JI) based on the homolog ID assigned to the gene in PLAZA was calculated for each stress. In addition, the p-value was determined through a permutation test where the observed JI of homolog IDs in a stress for each species was compared to the permuted JI of homolog IDs 1000 times.'

Discussion: *On the other hand, we did observe a significant conservation in differential gene expression between Arabidopsis and Marchantia gene families in cold, heat and salt, but no dark (Figure 5D). The latter result agrees with studies showing conservation of stress response in plants focusing on transcription factors and hormones*⁵⁷⁻⁶¹.

3.3 I do not think that the claim that the stress responses in *Arabidopsis* and *Marchantia* are different supports the idea that the knowledge obtained in *Arabidopsis* cannot be used to be transferred to crops (lines 819-822). Phylogenetically and morphologically, *Arabidopsis* and crops are more closely related than *Arabidopsis* and *Marchantia*.

Response: We thank the reviewer for highlighting this and have re-wrote the part:

'This suggests that each model plant uses a different strategy to acclimate to stress, and considerations regarding the phylogenetic distance between model species and crops of interest should be accounted for when attempting to make knowledge transfers from a model plant to crops.'

3.4 The authors could explain better what are the union GRNs in Figure S9B.

Response: We have included the following to explain union GRNs in the legend of Figure S9B:

'Union networks were generated by selecting for interactions found in at least X networks as indicated in the brackets. For example, 'Union (2)' will indicate that the interactions were found in at least 2 of the stress-specific networks.'

3.5 It would have been nice to have some functional validation of the GRNs in *Marchantia*, given its enormous potential as model system (carefully explained in the introduction).

Response: We fully agree with the reviewer that functional validation would have been nice. To this end, we have looked up all available publications of the 95 robustly-responding transcription factors. Perhaps not surprisingly, the majority of them have not been studied yet, and the few studies describe their roles in development, rather than stress. We describe these findings in text with:

'To study the roles of these 75 robustly-responding TFs, we considered the available literature on NCBI for studies that investigate the molecular function of the TFs. Out of the 75 TFs, 14 were found in literature and only 3 have been studied for their role in abiotic stress response (MpLAXR (Mp5g06970.1, robustly upregulated in high light and mannitol stress) and MpERF15 (Mp7g09350.1, robustly downregulated in heat stress): regeneration after wounding, MpHSF1 (Mp4g12230.1, robustly downregulated in heat stress): heat tolerance) (Table S7, Figure 3C). The remaining 11 transcription factors have roles in various developmental processes (e.g., gemma cup formation, Table S7), indicating that abiotic stress influences the development of Marchantia.'

Minor points:

3.6 Even though *Marchantia* is considered to have a general low gene redundancy compared to

Arabidopsis, this is mostly true for signaling components but not for secondary metabolism. Hence, the fact that Marchantia has several PALs is not entirely surprising. Please keep in mind that the number of Marchantia genes is roughly two thirds of the total number of Arabidopsis genes.

Response: We agree with the reviewer that signalling components can be highly duplicated even in a genome with low redundancy and have modified the statement as our phrasing was misleading:

'Marchantia contains a surprisingly large number (13) of PALs when compared to Arabidopsis (4).'

3.7 There is a lot of variability in the size of control plants (Fig. S1). Moreover, the 50 % growth inhibition of plants grown for 3 days in dark conditions is difficult to appreciate in Fig. 1B. Is this phenotype reproducible?

Response: We agree that there is variability in the size of control plants across batches. The significance is present in Figure 1 D and E when compared to all controls and also present when compared to its own batch control (Figure S1, Day 15 snapshot below). The 50% decrease in size is indeed difficult to observe in Figure 1B as the images here are of plants at Day 21. On the plot in Figure S1 (snapshot of Day 21 below), the plants have had the chance to recover and catch up to the size of the controls by day 21, although still significantly smaller. For better appreciation of the reproducibility of the phenotypes, we have now included Supplementary Data 1 that shows the original plates on day 15 and 21 across all experiments.

In addition, to account for the variability between batch controls, we have thus sequenced triplicates of 2 batch controls and applied a stringent cutoff of being significantly different from both batch controls in our differential gene expression analysis to be considered for downstream analysis. Moreover, the differential gene expression between batch controls was low between controls, as shown in Table S5, row 7, where only a total of 4% of genes were significantly differentially expressed.

3.8 Line 405: Please, check the references to the figures

Response: We thank the reviewer for spotting this, and have amended the references.

'Similarly, we observed core responses to heat (e.g., 6th and 10th column, Figure S5A) and cold (11, 12, 19 and 39 th column, Figure S5A).'

3.9 Line 500: Not in bottom row of S10, please correct

Response: We have updated the text:

'Figure 3B, *bottom row* and S10, *26th row from bottom*).'

3.10 TFs families in Figure 3 and S10 should have the same colors for clarity.

Response: We have updated the figure to match the colors of TF families in Figure 3 and S10

We would like to thank the reviewer for these great, spot-on comments!

Reviewer #1 (Remarks to the Author):

What are the noteworthy results?

See prior review response. This reviewer has no additional comments.

Will the work be of significance to the field and related fields? How does it compare to the established literature? If the work is not original, please provide relevant references.

See prior review response. This reviewer has no additional comments.

Does the work support the conclusions and claims, or is additional evidence needed?

The authors have strengthened the manuscript by providing further description and refinement of the analysis and methodology.

Are there any flaws in the data analysis, interpretation and conclusions? Do these prohibit publication or require revision?

The authors have adequately addressed the major concerns mentioned in the initial review. This includes further clarification of images and image captions, further explanation of model construction, and improved clarity of the explanation of results.

Is the methodology sound? Does the work meet the expected standards in your field?

The authors have adequately described the pros and cons of the modeling approach applied. The authors have also provided a more complete description of the approach, which is particularly helpful for the reader.

Is there enough detail provided in the methods for the work to be reproduced?

The authors have adequately updated the descriptions of the computational approaches used in the manuscript. This includes the modeling approach, clustering methods, and regression models that were presented. The authors have also made corrections of minor typos in the manuscript.

Additional Comments:

Issues associated with the additional comments in the prior review were adequately addressed.

Reviewer #2 (Remarks to the Author):

The manuscript has been greatly improved, but I still find some points described below.

- Lines 508-409, does 'with the majority of functions not being involved in heat stress' really refer to Figure 5B? Or I do not understand how to read Figure 5B, because there is no biological functions indicated in Figure 5B. Also, the labels for the x-axis in Figure 5A have been removed for some reason.

- In Table S9, some of the conditions in Arabidopsis appear to be duplicated. Explanation or correction needed.

Minor points

- Line 268, '(four stresses: H, HD, HS, HN)' should be '(four stresses: H, HM, HS, HN)'.

- Lines 271-272, '(bin not annotated)' -> '(bin 'not annotated')' for clarity.

- Lines 277-278, Consider adding abbreviations for the given combinations, e.g., 'salt+mannitol (SM)'.

- Lines 333-335, 'Mp' in the gene names should be in regular face, instead of italic.

- Lines 359-360, Figure S11B does not appear to directly show the statement 'most (89 out of 95) TFs are still connected to other TFs in the GRN'.
- Lines 450-457, labels 'Figure 6C' and 'Figure 6D' appear to be switched.
- Line 465, 'the parameters S_x of S_y values' could be 'the S_x and S_y parameters' ?
- Line 556, 'the acute stresses' might be 'the chronic stresses' ?
- Lines 1132-1133, I can't find green and red edges in Figure 4A?
- Fig. 2C, 'Uupregulated' -> 'Upregulated'. It might be clearer if a sentence like the following would be added to the description for Figs. 2C and D: 'The values were calculated from the equations given in B.'
- Table S10, 'no hits & (original description: none)' in the description column may be just 'no hits'?

Marchantia eFP Browser

This cool browser needs one more small correction: archegoniophores and antheridiophore of female and male plants, respectively, appear to be coming out from wrong places (my apologies, I should have noticed and pointed out in the previous review). They are formed from meristems at the terminal of a midrib. The following old drawings should help:

(female) <https://www.marchantia.org/new-gallery/m5mkuzpepv0chgeie4n6o46x7cw8nw>

(male) <https://www.marchantia.org/new-gallery/57r3hiuuicrkcjda2zqtoywhlcj377>

Reviewer #3 (Remarks to the Author):

In the revised version of the manuscript, the authors have addressed all the points raised by the reviewers. The current version of the manuscript has notably improved and is much easier to follow. At this point, I only have two minor suggestions:

Line 251: Figure 2C-D should be referenced instead of 2E-F

Line 495: It is worth mentioning that eFP includes the expression data on the single and combined abiotic stresses (not just the anatomy of the plant) and that MBEX and CoNeKT do not include the same expression datasets.

REVIEWERS' COMMENTS

Reviewer #1 (Remarks to the Author):

What are the noteworthy results?

See prior review response. This reviewer has no additional comments.

Will the work be of significance to the field and related fields? How does it compare to the established literature? If the work is not original, please provide relevant references.

See prior review response. This reviewer has no additional comments.

Does the work support the conclusions and claims, or is additional evidence needed?

The authors have strengthened the manuscript by providing further description and refinement of the analysis and methodology.

Are there any flaws in the data analysis, interpretation and conclusions? Do these prohibit publication or require revision?

The authors have adequately addressed the major concerns mentioned in the initial review. This includes further clarification of images and image captions, further explanation of model construction, and improved clarity of the explanation of results.

Is the methodology sound? Does the work meet the expected standards in your field?

The authors have adequately described the pros and cons of the modeling approach applied. The authors have also provided a more complete description of the approach, which is particularly helpful for the reader.

Is there enough detail provided in the methods for the work to be reproduced?

The authors have adequately updated the descriptions of the computational approaches used in the manuscript. This includes the modeling approach, clustering methods, and regression models that were presented. The authors have also made corrections of minor typos in the manuscript.

Additional Comments:

Issues associated with the additional comments in the prior review were adequately addressed.

We thank the reviewer for the positive comments.

Reviewer #2 (Remarks to the Author):

The manuscript has been greatly improved, but I still find some points described below.

3.1

- Lines 508-409, does 'with the majority of functions not being involved in heat stress' really refer to Figure 5B? Or I do not understand how to read Figure 5B, because there is no biological functions indicated in Figure 5B. Also, the labels for the x-axis in Figure 5A have been removed for some reason.

Text elaborated to: 'the majority of functions not being involved in heat stress (Figure 5B, <20% Arabidopsis orthologs involved in heat response, first column on the left)'. The figure has been fixed.

3.2

- In Table S9, some of the conditions in Arabidopsis appear to be duplicated. Explanation or correction needed.

We are sorry for the error. The bug has been fixed and the table for the counts are updated.

3.3

Minor points

- Line 268, '(four stresses: H, HD, HS, HN)' should be '(four stresses: H, HM, HS, HN)'.

Corrected, thank you.

3.4

- Lines 271-272, '(bin not annotated)' -> '(bin 'not annotated')' for clarity.

Corrected, thank you.

3.5

- Lines 277-278, Consider adding abbreviations for the given combinations, e.g., 'salt+mannitol (SM)'.

Corrected, thank you.

3.6

- Lines 333-335, 'Mp' in the gene names should be in regular face, instead of italic.

Corrected, thank you.

3.7

- Lines 359-360, Figure S11B does not appear to directly show the statement 'most (89 out of 95) TFs are still connected to other TFs in the GRN'.

most (89 out of 95) TFs are still connected to other TFs in the GRN (Figure S11A, red line, Figure S11B).

Corrected, thank you.

3.8

- Lines 450-457, labels 'Figure 6C' and 'Figure 6D' appear to be switched.

Corrected, thank you.

3.9

- Line 465, 'the parameters Sx of Sy values' could be 'the Sx and Sy parameters' ?

Corrected, thank you.

3.10

- Line 556, 'the acute stresses' might be 'the chronic stresses' ?

No, correct as is.

3.11

- Lines 1132-1133, I can't find green and red edges in Figure 4A?

Corrected, thank you.

3.12

- Fig. 2C, 'Uupregulated' -> 'Upregulated'. It might be clearer if a sentence like the following would be added to the description for Figs. 2C and D: 'The values were calculated from the equations given in B.'

Corrected, thank you.

3.13

- Table S10, 'no hits & (original description: none)' in the description column may be just 'no hits'?
Fixed

Corrected, thank you.

3.14

Marchantia eFP Browser

This cool browser needs one more small correction: archegoniophores and antheridiophore of female and male plants, respectively, appear to be coming out from wrong places (my apologies, I should have noticed and pointed out in the previous review). They are formed from meristems at the terminal of a midrib. The following old drawings should help:

(female) <https://www.marchantia.org/new-gallery/m5mkuzpepv0chgeie4n6o46x7cw8nw>

(male) <https://www.marchantia.org/new-gallery/57r3hiuicrkcjda2zqtowywhlcj377>

We have updated the eFP browser.

Reviewer #3 (Remarks to the Author):

In the revised version of the manuscript, the authors have addressed all the points raised by the reviewers. The current version of the manuscript has notably improved and is much easier to follow. At this point, I only have two minor suggestions:

4.1

Line 251: Figure 2C-D should be referenced instead of 2E-F

Corrected, thank you.

4.2

Line 495: It is worth mentioning that eFP includes the expression data on the single and combined abiotic stresses (not just the anatomy of the plant) and that MBEX and CoNeKT do not include the same expression datasets.

Corrected, thank you.